# Acceleration of dolomitization by zinc in saline waters

Veerle Vandeginste [1,2], Oliver Snell[1], Matthew R. Hall[2,3], Elisabeth Steer[4] & Arne Vandeginste[5]

Dolomite ($CaMg(CO_3)_2$) plays a key role in the global carbon cycle. Yet, the chemical mechanisms that catalyze its formation remain an enigma. Here, using batch reactor experiments, we demonstrate an unexpected acceleration of dolomite formation by zinc in saline fluids, reflecting a not uncommon spatial association of dolomite with Mississippi Valley-type ores. The acceleration correlates with dissolved zinc concentration, irrespective of the zinc source tested ($ZnCl_2$ and $ZnO$). Moreover, the addition of dissolved zinc counteracts the inhibiting effect of dissolved sulfate on dolomite formation. Integration with previous studies enables us to develop an understanding of the dolomitization pathway. Our findings suggest that the fluids' high ionic strength and zinc complexation facilitate magnesium ion dehydration, resulting in a dramatic decrease in induction time. This study establishes a previously unrecognized role of zinc in dolomite formation, and may help explain the changes in dolomite abundance through geological time.

[1] School of Chemistry, University of Nottingham, University Park, NG7 2RD Nottingham, UK. [2] GeoEnergy Research Centre, University of Nottingham, University Park, NG7 2RD Nottingham, UK. [3] British Geological Survey, Environmental Science Centre, Keyworth, NG12 5GG Nottingham, UK. [4] Nano- and Microscale Research Centre, University of Nottingham, University Park, NG7 2RD Nottingham, UK. [5] Camco Technologies, Haasrode Research Park 1040, Technologielaan 13, 3001 Leuven, Belgium. Correspondence and requests for materials should be addressed to V.V. (email: veerle.vandeginste@nottingham.ac.uk)

Global warming, triggered by the rapid increase of anthropogenic carbon dioxide and other greenhouse gas emissions, is a major concern and calls for the need to better understand the global carbon cycle[1,2]. The carbonate reservoir of the Earth's crust accounts for the major component of global carbon storage. Dolomite $(CaMg(CO_3)_2)$ stores 14% more carbon per unit volume, and is less prone to chemical weathering, than the only carbonate mineral that is more abundant on Earth, calcite $(CaCO_3)$. The rare occurrence of dolomite in recent marine sedimentary systems (despite modern oceans being supersaturated with respect to dolomite) in contrast to its abundance in ancient sedimentary systems remains an enigma[3,4]. Dolomite can be found, for example, in hypersaline lagoon sedimentary environments and in burial settings where dolomite may spatially occur alongside ore deposits. It has proven to be extremely challenging to form crystallographically well-ordered dolomite in the laboratory under ambient conditions due to the slow reaction kinetics[5]. The unfavorable kinetics have been attributed mainly to the strong hydration enthalpy of magnesium ions[6], the large energy barrier for long range ordering of $Mg^{2+}$ and $CO_3^{2-}$ ions in the crystal[7], and lattice stress buildup linked to the impurity incorporation model[8,9] and self-limiting growth model[10,11]. Despite the lack of convincing evidence of successful laboratory synthesis of dolomite at ambient temperature[12], experimental studies demonstrate that the formation of protodolomite, a precursor of dolomite, is favored by dissolved sulfide[13] in contrast to dissolved sulfate under certain conditions[14,15] (debated by some authors[6,16,17]), by high salinity[14], microbial mediation such as sulfate-reducing bacteria[15,18], extracellular polymeric substances[19,20], and carboxylated surfaces on organic matter[21]. Despite the research conducted to date, the chemical pathways behind the catalysis of dolomitization (calcite-to-dolomite replacement) remain poorly understood.

Previous research investigating factors affecting the kinetics of dolomite formation predominantly built on the common association of dolomite with salt lakes and lagoons[14]. In addition to those environments, dolomite is also often the host rock of zinc ores of the Mississippi Valley-type (MVT)[22–24]. Interestingly, zinc ions have a hydration enthalpy stronger than that of magnesium ions despite a similar ionic radius[25]. Moreover, zinc drives mineralization leading to the formation of kidney stones[26] or bladder stones, documented to contain dolomite in some cases[27]. Furthermore, carbonic anhydrase (CA), a Zn-containing metalloenzyme, accelerates the formation of calcium carbonate, for example in sponge spicules[28], and in alkaline solutions[29]. The role of zinc in each of these earth science, chemistry, and biology contexts is striking; yet, no previous research has investigated the impact of dissolved zinc on the reaction pathways and kinetics of dolomitization.

Here, we demonstrate for the first time that a radically different, previously rarely explored effector—namely zinc ions in saline solutions—acts as an efficient catalyst for dolomitization. Moreover, the experimental data show that the catalytic effect of zinc counteracts the inhibiting effect of sulfate on dolomite formation. Our findings may help explain the common spatial association of dolomite with MVT ore formation, establish a previously unrecognized link between the geochemical cycles of carbon and zinc with the involvement of abiotic carbonate formation, and may increase our understanding of dolomite abundance through geological time.

## Results

**Batch reactor experiments**. We postulate that the competitive binding affinity of Zn with a range of ligands to form complexes and its flexible coordination (4, 5, and 6), in comparison with Mg (and Ca) ions, enhances $Mg^{2+}$ dehydration, facilitating incorporation of Mg in the crystal structure, and thus resulting in a catalytic effect of Zn ions on dolomite formation from Zn-bearing solutions. To test this hypothesis, we conducted calcite-to-dolomite replacement batch reactor experiments at 200 °C in which 200 mg of calcite was reacted with 15 ml of simulated natural brines containing 2.00 M NaCl, 0.30 M $MgCl_2$ and 0.20 M $CaCl_2$ in the control series, and added $ZnCl_2$ (0.01, 0.03, 0.05, 0.10, and 0.20 M) in fluids of the same ionic strength of 3.5 (by reducing the concentration of NaCl) in the $ZnCl_2$ experiments.

**Reaction pathway**. Detailed powder X-ray diffraction (PXRD) analysis of the reaction products shows that the dolomitization reaction occurs in three stages: first an induction period; followed by a conversion stage of the reactant, Mg-poor calcite, to protodolomite; and finally a recrystallization stage of protodolomite to well-ordered dolomite. This transition from calcite to protodolomite is detected by the shift in the (104) peak (Fig. 1), whereas protodolomite and dolomite are distinguished by an ordering (015) peak (Fig. 2). Besides calcite, protodolomite and dolomite, we have identified traces of brucite, $Mg(OH)_2$, and simonkolleite, $Zn_5(OH)_8Cl_2\bullet(H_2O)$, in the reaction products from the control series and the $ZnCl_2$ experiments, respectively. The presence of simonkolleite in the reaction products is even more pronounced in an additional experimental series where zincite (ZnO) was used instead of $ZnCl_2$ (Fig. 3). The corresponding geochemical elemental data imply that zincite dissolves in the saline solution at 200 °C within the first 3 h (Fig. 3). Moreover, since the Zn concentration in the reaction products lacks a distinct correlation with (proto)dolomite content (Supplementary Fig. 1), it is interpreted to be caused by the presence of simonkolleite, rather than significant incorporation of Zn in the (proto)dolomite structure. Hydrothermal dolomite hosting MVT deposits may contain ppm to % contents of Zn, and zincian dolomite has been considered a peripheral hydrothermal hypogene alteration product in Polish ore deposits[30], and a low temperature supergene alteration product of primary sulfide deposits in Jabali (Yemen), Iglesias (SW Sardinia), and Yanque (Peru)[31].

Many replacement reactions have been shown to involve interface-coupled dissolution-precipitation[32,33], which is probably the key pathway in the second stage of the dolomitization reaction. With abundant $Ca^{2+}$ and $Mg^{2+}$ available in the solutions by addition of soluble chloride salts but no dissolved carbonate, some calcite must dissolve to enable new carbonate formation. Hence, calcite dissolution and the formation of protodolomite are intimately coupled and must take place at the fluid-mineral interface. The new crystals have been reported to grow epitaxially onto the dissolving calcite surface[33], in contrast to spherulitic crystal growth, as documented in direct dolomite precipitation experiments[34]. The third stage involves Ostwald ripening dissolution-reprecipitation during transformation of protodolomite to dolomite. The dolomite product from this stage is crystallographically well-ordered and stoichiometric (50 mole% $MgCO_3$) in each of the experiments, thus irrespective of the presence of zinc or sulfate ions in the solution. The stoichiometry of the dolomite precipitates is likely favored by the Mg/Ca ratio of 1.50, consistent with the findings by Kaczmarek et al.[35]. These dolomite crystals have a rhombic shape and sub-micron diameter (about 0.3 to 0.8 µm for reaction product of 14 h in the main Zn experiment series; Supplementary Fig. 2). This contrasts with ellipsoidal, rice-grain shaped calcite particles in the first stage of the reaction process (Supplementary Fig. 2). The

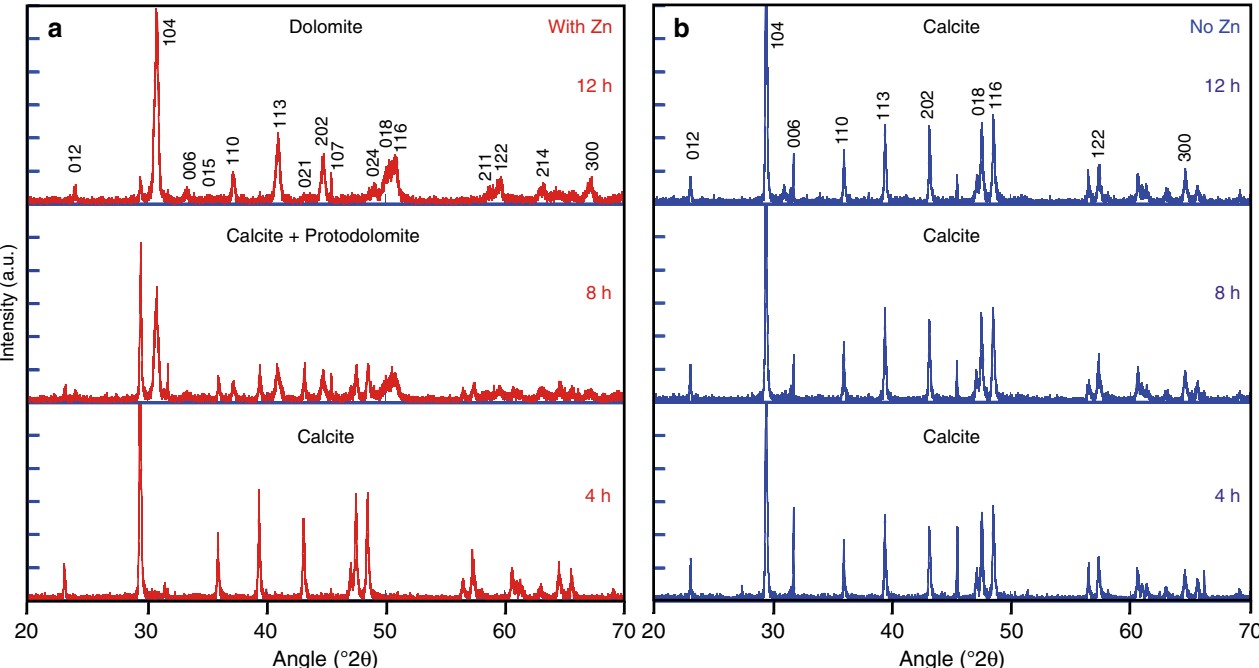

**Fig. 1** Powder X-ray diffractograms (selected 20–70 °2θ) of reaction products in the main experiment series at selected time intervals of 4, 8, and 12 h. The dolomite and calcite characteristic reflections are indicated in **a** and **b**, respectively. The peak at 31.7 °2θ in some diffractograms is due to minor halite that was not entirely dissolved during the rinsing process of the product or developed during drying of the samples

small size of the dolomite crystallites stems from fast nucleation and growth of the protodolomite precursor.

**Effect of zinc on dolomitization rate**. We found that the calcite-to-dolomite replacement reaction is accelerated by the addition of dissolved Zn to NaCl–MgCl$_2$–CaCl$_2$ aqueous solutions of constant ionic strength, with greater acceleration for higher Zn concentration within the range tested (Fig. 4; Supplementary Tables 1–3). The main experiment series, with 0.2 weight% Zn, simulates natural fluids reported for the formation of carbonate-hosted Zn sulfides[23]. The results reveal a significant decrease in induction time, and an increase in average reaction rate of protodolomite formation, by the addition of Zn in solution (Fig. 4; Supplementary Tables 2–3). The kinetics for such mineral replacement reaction can be modeled by the Avrami equation[5,34,36,37], which states that the reaction extent $y = 1 - \exp\left[-(k(t - t_0))^n\right]$ whereby $k$ is the rate constant, $t$ is the reaction time and $t_0$ the induction time, and $n$ is a time exponent that depends on the reaction mechanism[38]. The best fit for the Avrami model indicates an induction time of 10.5 h, a rate constant of $3.9 \times 10^{-5}\,s^{-1}$, and time exponent of 2.4 for the control dolomitization reaction at 200 °C, and a lower induction time of 4.5 h, a higher rate constant of $7.6 \times 10^{-5}\,s^{-1}$, and lower time exponent of 1.6 for the dolomitization experiments with ZnCl$_2$ (Fig. 5). The geochemical and mineralogical data show that the dolomitization reaction rate in the ZnO experiments is very similar to that in the ZnCl$_2$ experiments, supporting the hypothesis of the accelerating impact of Zn ions (Fig. 3, Supplementary Tables 3–8). The Avrami model for ZnO experiments indicates very similar data as for the ZnCl$_2$ experiments, and shows an induction time of 4.5 h, a rate constant of $7.2 \times 10^{-5}\,s^{-1}$ and time exponent of 1.8 (Fig. 5). In additional experiments with sulfate, we have verified that also in the presence of dissolved sulfate, which may inhibit dolomite formation under certain conditions[14,15], Zn still shows a significant accelerating impact on the dolomitization rate (Supplementary Fig. 3, Supplementary Tables 9–10).

## Discussion

The rate of the reaction of calcite to protodolomite is controlled by the slowest of the following three processes: transport of solutes to and from the reaction front; calcite dissolution; and protodolomite precipitation. Because of the high diffusion rate at 200 °C and fast calcite dissolution[39,40], protodolomite formation is the rate-limiting factor. Addition of ZnCl$_2$ in the fluids decreases the pH from 8.6 to 6.7 (measured at 25 °C), caused by the formation of Zn(II) aquo complexes[41]. A lower pH is not likely to accelerate protodolomite precipitation, however, the formation of the complexes is. Among the divalent ions, Mg forms one of the strongest bonds with water molecules resulting in the [Mg(H$_2$O)$_6$]$^{2+}$ complex, and this strong hydration status of Mg ions is accepted as a main inhibitor in rapid dolomite formation from aqueous fluids[13]. Molecular dynamic simulations have shown that once Mg ions are adsorbed onto the mineral surface, they are inhibited from diffusing into the bulk lattice by water molecules[42]. The addition of NaCl in the fluids increases the dolomite formation rate (similar to addition of LiCl in the study by Gaines[43]), because Mg-water complexes are less stable in fluids of higher ionic strength, following the Debye-Hückel theory and extensions. This facilitates Mg dehydration for dolomite crystal formation and growth, because many water molecules form hydration shells around the Na and other ions in solution. For the solutions in our experiments, we calculate that there are only about 10 molecules of water per ion. Given that Mg can form complexes with 6 water molecules in the first hydration shell and 12 in the second hydration shell, there may be a competitive effect in ion-water complexation. Our findings demonstrate that the presence of Zn ions, which have a stronger hydration enthalpy than Mg ions, appear to promote Mg dehydration in aqueous solutions, whilst favouring incorporation of Mg into the proto-dolomite structure. Zinc forms strong complexes with water and several anions, and has a flexible coordination between 4, 5 and 6, which makes it an efficient catalyst in biological context[41]. We propose that the competitive binding affinity of Zn (in comparison with other metals) with water, carbonate, and chloride[44] (all

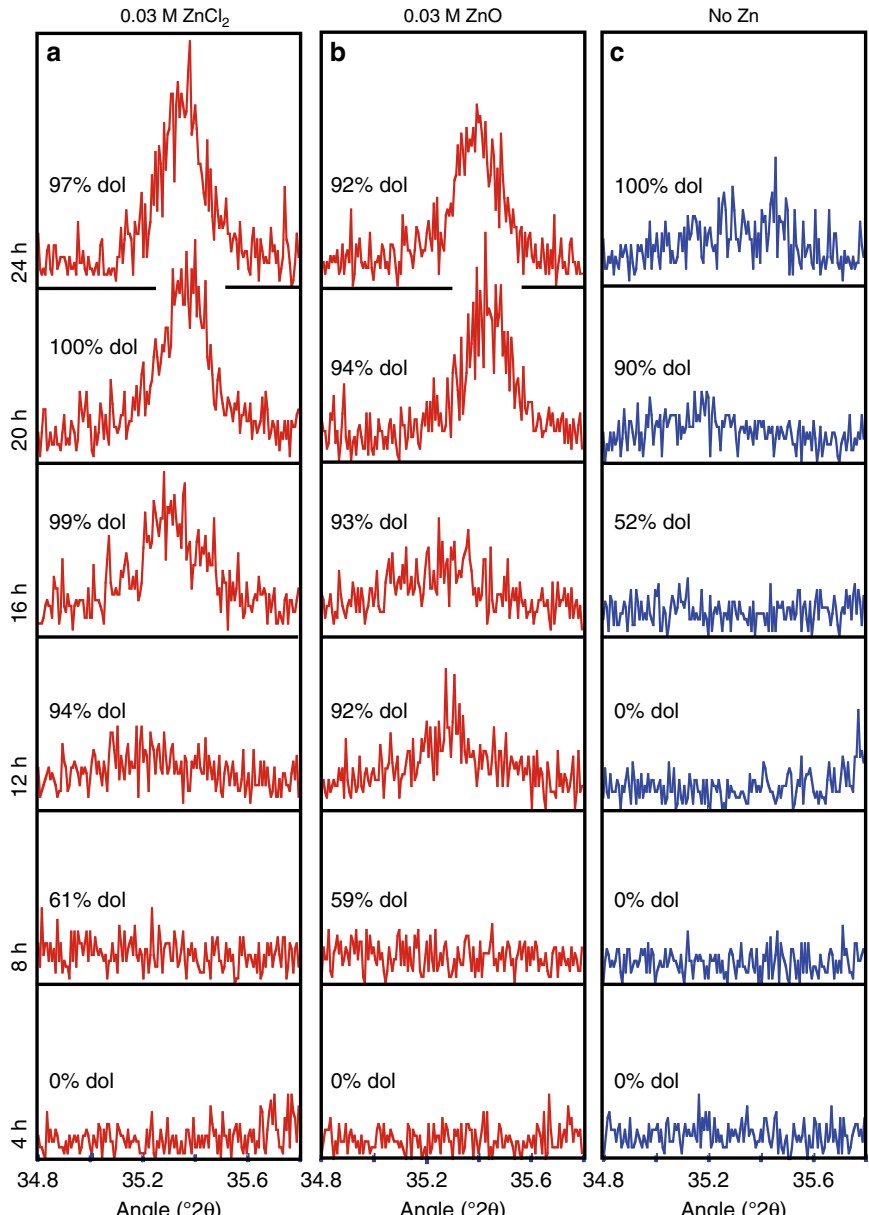

**Fig. 2** Powder X-ray diffractogram selected area (34.8-35.8 °2θ) of reaction products. The crystallographic data of the reaction products in the 0.03 M ZnCl₂ experiments (**a**), the 0.03 M ZnO experiments (**b**), and the Zn-free control series (**c**) show that only reaction products with more than 90% dolomite contain well-ordered dolomite, giving an indication of the protodolomite to dolomite recrystallization rate. The % dol refers to the percentage of (proto)dolomite as calculated from the main (104) dolomite peak

present in the solutions), and its easy interconversion between coordination may facilitate removal of water molecules from Mg-water complexes at the crystal interface.

Fluids play an essential role in the mobilization of chemical elements; and fluid-rock interaction, common in the burial environment, impacts on the cycling of elements. Many MVT ore bodies are surrounded by a halo of hydrothermal dolomite[45]. The mineralization event, which may comprise several stages of dolomite and sulfide ore formation, involves a hydrothermal system[46] with metal-rich brines, whereby either mixing with a second, sulfide-rich fluid, or in situ sulfate reduction leads to precipitation of sulfide ores[47]. Hydro-thermal metal-rich saline fluids may have caused also dolomiti-zation in these environments, and may help explain why MVT deposits are more commonly associated with dolostone than limestone[48].

The importance of Zn in accelerating carbonate formation is also identified in marine organisms, where CA, a Zn-containing enzyme, regulates biogenic carbonate formation, as documented in corals[49], sponges[28], and other metazoans[50]. The catalytic mechanism of CA involves the attack of Zn-bound $OH^-$ on a $CO_2$ molecule loosely bound in the active center of the enzyme, and the resulting Zn-coordinated $HCO_3^-$ ion is then displaced from the metal ion by $H_2O$. Zinc also stimulates the production of extracellular polymeric substances[51], which have been shown to catalyse dolomitization[20]. Zinc is an essential trace nutrient for marine microorganisms, as demonstrated by the typical nutrient cycling profile of Zn concentrations in the ocean, with surface depletion due to phytoplankton uptake and regeneration at depth by remineralization of sinking organic matter. Uptake of $HCO_3^-$ by marine diatoms is controlled by the concentrations of $CO_2$ and of inorganic Zn, suggesting that the availability of Zn may limit

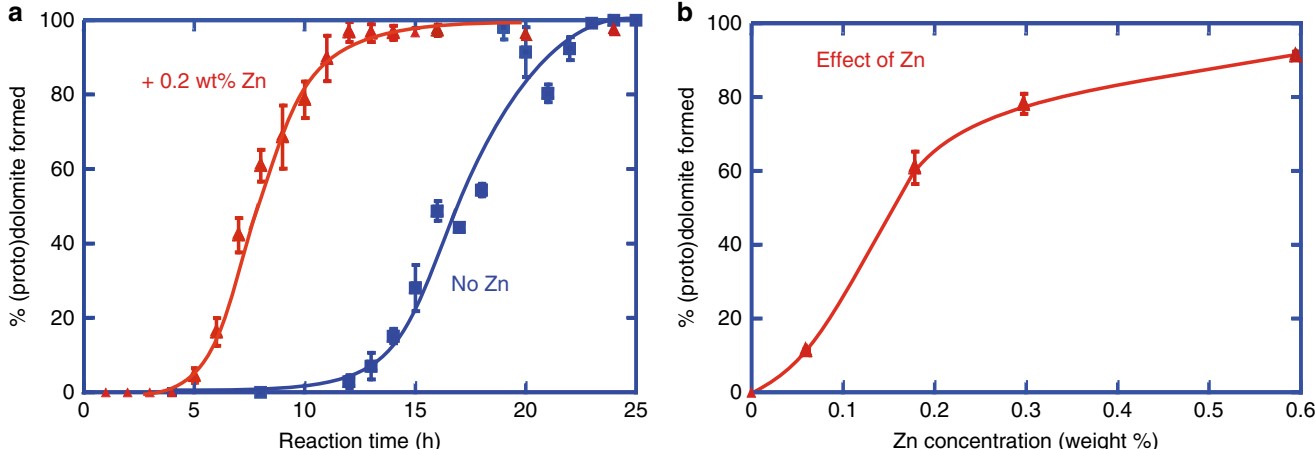

**Fig. 3** Plots of geochemical and mineralogical data of reaction products against reaction time. The geochemical major element concentrations for the reaction products in the 0.03 M $ZnCl_2$ experiments (**a**) and in the 0.03 M ZnO experiments (**b**), as well as the relative mineral abundance for the reaction products in the 0.03 M $ZnCl_2$ experiments (**c**) and in the 0.03 M ZnO experiments (**d**) demonstrate very similar dolomitization reaction rates which are both accelerated compared to Zn-free solutions. The ZnO (zincite) experiments show dissolution of zincite within the first three hours of the experiment and significantly larger amount of simonkolleite formed in the reaction (**d**) compared to the $ZnCl_2$ experiments (**c**)

**Fig. 4** Calcite-to-dolomite replacement reaction curves. **a** Reaction curve showing that the addition of 0.2 weight% zinc in the fluids (red curve in comparison with blue curve) halves the time required for dolomitization of calcite at 200 °C. **b** Small amounts of zinc ions added to solutions of the same ionic strength increase significantly the amount of (proto)dolomite formed within 8 h reaction time. Error bars represent standard deviation of replicate samples

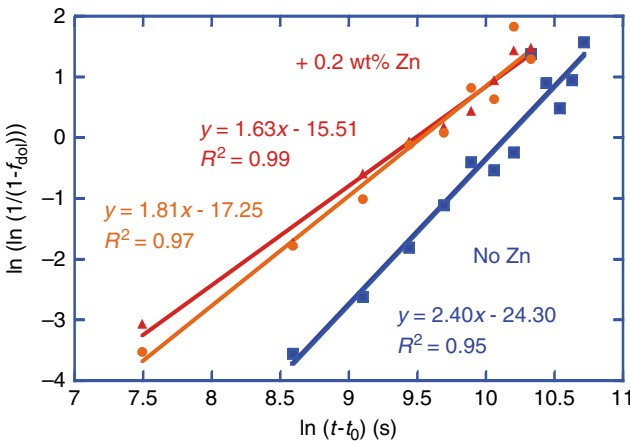

**Fig. 5** Avrami plot for dolomitization of calcite at 200 °C. The linear best fit trend lines enable derivation of time exponent and rate constant for the control dolomitization experiments (blue), ZnCl₂ dolomitization experiments (red), and ZnO dolomitization experiments (orange)

oceanic phytoplankton production and influence the global carbon cycle[52]. Moreover, the recently identified strong Zn-silicate linear relationship in ocean water (with vertical Zn and Si profiles showing nutrient-like distributions) demonstrates zinc's control on the efficiency of the global biological carbon pump[53]. This link between the geochemical Zn cycle and the global carbon cycle is even more important than previously thought, based on our discovery of the accelerating effect of Zn ions on abiotic carbonate formation in saline waters. Periods of low atmospheric oxygen and widespread marine anoxia in past climates have been favorable for the formation of abiotic authigenic carbonates in marine sediment pore fluids, representing a significant carbon sink[54]. Moreover, the contribution of oceanic carbonate production by marine fish[55] is linked to the biological evolution through geologic time. The variation in atmospheric, biological, and geochemical conditions implies that also the type of authigenic carbonate (calcite, aragonite, dolomite) formed may have been strongly controlled, and their formation rate significantly catalyzed, by Zn concentrations in seawater derived fluids. An example of such climatic effect has been demonstrated in the accumulation of dolomite in coralline algae on tropical reefs[56].

In conclusion, we have documented the reaction pathways and rates of calcite-to-dolomite replacement in a series of saline solutions. Our results show that the reaction rate is significantly higher in the presence of Zn ions. The experiments also demonstrate that Zn ions counteract the inhibiting effect of sulfate ions on dolomite formation (under the temperature conditions of the experiments). These observations suggest that the presence of dissolved Zn in saline fluids favors the dehydration of Mg, thus enhancing its availability for incorporation in the dolomite structure. Our findings establish a previously unrecognized link between geochemical cycling (of Zn, Mg, Ca, and C) as part of mass transfer (geochemical element exchanges) between fluids and rocks in the Earth's crust. The dolomitization experiments in this study were conducted with fluid compositions and temperature conditions that may help explain the common association of dolomitization with MVT Zn ore formation. Whether these findings can be extrapolated to low temperature conditions is still a question. Future research will involve testing the effect of dissolved Zn on the rate of dolomitization at low temperature. Those results will then allow extending our findings across temperature scale and spatial regions. Moreover, these would enable drawing more inferences on geochemical cycling in

surface sedimentary environments. This new understanding may also help explain the intriguing changes in dolomite abundance through geological time.

## Methods

**Batch reactor experiments.** CaCO₃ (200 mg) was reacted with 15 ml of saline solution, as specified below, in teflon-lined steel batch reactors at 200 °C. A temperature of 200 °C was selected for the experiments based on the dual reasoning of both having relatively short reaction times, enabled by high temperature, to generate dolomite at favorable time scales for lab experimental work to test the catalytic impact of zinc, and, because this temperature falls within the temperature range at which Mississippi-Valley type zinc-lead ore deposits are formed.

Two experiment series were conducted whereby samples were taken every 4h over a total time of 56 h, with the Zn-free control saline solution, composed of 2.00 M NaCl, 0.30 M MgCl₂ and 0.20 M CaCl₂, and the Zn-bearing saline solution, composed of 1.85 M NaCl, 0.30 M MgCl₂ and 0.20 M CaCl₂, and 0.05 M ZnCl₂.

Five duplicate experiment series were conducted for a total reaction time of 8 h with solutions with varying dissolved zinc content, but all the same ionic strength of 3.5: 1.97 M NaCl, 0.30 M MgCl₂ and 0.20 M CaCl₂, and 0.01 M ZnCl₂; 1.91 M NaCl, 0.30 M MgCl₂ and 0.20 M CaCl₂, and 0.03 M ZnCl₂; 1.85 M NaCl, 0.30 M MgCl₂ and 0.20 M CaCl₂, and 0.05 M ZnCl₂; 1.70 M NaCl, 0.30 M MgCl₂ and 0.20 M CaCl₂, and 0.10 M ZnCl₂; 1.40 M NaCl, 0.30 M MgCl₂ and 0.20 M CaCl₂, and 0.20 M ZnCl₂.

Two triplicate experiment series were run with sampling at 1 h time steps and with the following solutions: Zn-free saline (control) solution (with starting pH of 8.6), composed of 2.00 M NaCl, 0.30 M MgCl₂ and 0.20 M CaCl₂, in the first main series, and Zn-bearing saline solution (with starting pH of 6.7), composed of 1.91 M NaCl, 0.30 M MgCl₂ and 0.20 M CaCl₂, and 0.03 M ZnCl₂, in the second main series. The Zn concentration in the latter fluid is 0.2 weight%, similar to that reported in Irish-type Zn sulfides[16]. The fluids were all set to have the same ionic strength of 3.50.

The presence of traces of ZnO in the reaction products of the ZnCl₂ experiments triggered the set-up of an additional experiment series (with 1 h time step sampling) whereby 200 mg of calcite and 36.6 mg of ZnO was reacted with 15 ml of a saline solution composed of 2.00 M NaCl, 0.30 M MgCl₂, and 0.20 M CaCl₂. The amount of zinc in those experiments is equivalent to the amount used in the ZnCl₂ experiments. Furthermore, we tested the impact of Zn in the presence of sulfate, given the known inhibiting effect of sulfate on dolomite formation. Two experiments series were set up whereby 200 mg of calcite was reacted with 15 ml of solution: Zn-free saline (control) solution with sulfate, composed of 2.00 M NaCl, 0.27 M MgCl₂, 0.03 M MgSO₄, and 0.20 M CaCl₂, in the first main series, and Zn-bearing saline solution with sulfate, composed of 2.00 M NaCl, 0.30 M MgCl₂ and 0.20 M CaCl₂, and 0.03 M ZnSO₄, in the second main series. These reactions were conducted at 200 °C for a series of reaction times (12, 24, 30, 36, 48, 60, and 72 h). Calcite powder was purchased from Acros Organics, NaCl from VWR, CaCl₂ from Merck, MgCl₂, MgSO₄ and ZnSO₄.7H₂O from Alfa Aesar, ZnCl₂ from Sigma Aldrich and ZnO from Honeywell.

**Calcite reactant morphology and surface area analysis.** Individual calcite grains were determined to be submicrometer size based on scanning electron microscopy (SEM) and energy dispersive spectrometry (EDS) analysis. The SEM and EDS analyses were performed on a Quanta 650 with an 150 mm Oxford Instruments X-Max detector at 10–15 kV, spot 2.5–4. Triplicate BET analyses were performed on the calcite powders using Micromeritics 3Flex surface characterization and nitrogen gas. The calcite powder has a N₂ Brunauer-Emmett-Teller (BET) surface area of 8.72 ± 0.04 m² g⁻¹.

**Mineralogy and geochemistry of reaction products.** The reaction products from the batch reactor experiments were collected by transfer to centrifuge tubes and cleaned three times with milliQ water, decanting the liquid after centrifugation at 4500 rpm for 5 min. This cleaning procedure should dissolve and remove most of the NaCl salt from the products. The residue was subsequently dried in an oven at 60 °C. The weight of the reaction product was then recorded. Complete replacement of 200 mg calcite is expected to result in 184 mg dolomite. Identification of the reaction product, dolomite, was conducted by PXRD, EDS-SEM analysis and ICP-OES analysis. The reaction extent was derived from the calcite and dolomite abundance in the reaction products, and verified by the expected weight of the reaction products (based on mole-per-mole replacement reaction), which was consistent in all cases (with up to 5% error as derived from triplicate experiments).

For PXRD analysis, the samples were scanned over a sampling range of 5 to 70 °2θ, with a step size of 0.0066 and a scan speed of 0.023 °2θ per second. A PANalytical X'Pert Pro was used for these analyses with CuKα radiation at 40 kV and 40 mA. A 5 mm mask was used because of the relatively small size of the samples contained in custom-built holders. Calcite and minor halite peaks in the samples were used as internal reference to assess machine drift. The relative ratios of dolomite and calcite in the samples were calculated based on the ratio of respective main peak areas. The background noise was removed before calculating integrated peak areas for the minerals detected to determine relative ratios of mineral abundance.

For geochemical analysis of the reaction products, 10 mg of the precipitate was dissolved in 10 ml 5% $HNO_3$ in small closed teflon reactors at 80 °C for one hour. Upon cooling, the solutions were transferred to centrifuge tubes, and centrifuged at 4500 rpm for 5 min. The fluid samples were then measured using a Perkin Elmer spectrometer Optima 2000 DV. A standard solution containing 28 elements at 100 ppm (Fisher Chemicals) was used to make a calibration series of 0.1, 1, 3, 5, 7, and 10 ppm solutions, made with 2% $HNO_3$. Sample fluids were diluted as necessary to obtain concentrations falling well within the calibration range.

## Data availability

The quantitative data for each of the samples from this study as well as the Tables and Figures in the Supplementary Files are available in the EarthChem Library (https://doi.org/10.1594/IEDA/111274). Additional information related to this paper can be obtained from the corresponding author upon request.

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

## Acknowledgements

We thank William Lewis for maintenance of the powder X-ray diffractometer, Mark Guyler and Ben Pointer-Gleadhill for the ICP-OES in the School of Chemistry, and Aleksandra Gonciaruk for help with the BET analyses in the Gas Adsorption Analysis Suite lab in Engineering. The manuscript has significantly improved by discussions and review of drafts by Neil Champness, Martyn Poliakoff, Liam Ball, Hon Lam, Matthew Jones, Paul Nathaniel, David Large, Deborah Kays, and Jonathan McMaster. Funding: We acknowledge support by the Engineering and Physical Sciences Research Council (EPSRC) under grant no. EP/K005138/1 and grant no. EP/M000567/1, and the British Geological Survey and Strategic Development Fund for the GeoEnergy Research Centre.

## Author contributions

V.V. designed the research and carried out the experiments and analyses unless otherwise stated. O.S. carried out preliminary experimental tests. E.S. performed the SEM-EDS analyses. A.V. developed software for data analysis. V.V., O.S., M.R.H., E.S., A.V, contributed to interpretation of the results.

## Additional information

**Competing interests:** The authors declare no competing interests.

