## [Peer Review File · Nature Communications]

Reviewers' comments:

Reviewer #1 (Remarks to the Author):

Very interesting paper, with some inaccuracies, though (e.g. the fact that Zn dolomite is only associated with weathering: this is not true because several hydrothermal dolomites have Zn in their lattice). The experiments look exact to me, but I have still some problems in imagining a dolomitization process on an already hard limestone by this kind of fluids in such a short time as that envisaged by the AA.

The relationships between dolomitization (the word hydrothermal was NEVER used) and MVT deposits are, in my opinion, overinterpreted and based on cursory evidence only. Of course MVT are often contained in dolomitized lithotypes, but this is not enough to link them to the same fluids, especially if the two processes are taking place at different times.

The AA mention several times that their results are important because there is a known connection between MVT Zn-Pb deposits and dolomitization, but they never discuss this connection in detail: if the dolomitizing fluid (with Zn in solution) will also precipitate Zn sulfides afterwards, or what will happen to the Zn that was catalysing the dolomitization process?

If the AA. can at least hypothesize (and describe) the relationships between dolomitization and MVT ores (and not only their spatial association), this would be a great addition to the paper. Otherwise, this aspect should not be specifically mentioned.

Reviewer #2 (Remarks to the Author):

Review of “The role of zinc in dolomitization and the inorganic carbon cycle”, by Veerle Vandeginste, Oliver Snell, Matthew R. Hall, Elisabeth Steer, and Arne Vandeginste, submitted to *Nature Communications* (ms# NCOMMS-18-24268-T).

Summary and recommendation —

This paper describes the results of hydrothermal experiments (200°C), that show addition of ZnCl₂ in Na-Ca-Mg-Cl brines increases the net conversion of powdered calcite to dolomite relative to Zn-free control solutions of similar ionic strength. The primary data indicate that at these temperatures, this complex multi-step reaction is accelerated relative to their controls. All well and good. However, the authors’ experiments and treatment of the data derived from them supply little insight and no hard numbers of the kinetics of the process itself. There are no data to evaluate the rate of the process in conventional units of mineral kinetics (mass per unit area per unit time), no calculations as to the *in situ* conditions driving this process. Thus we can not place these results in the context of published (but uncited here) rate laws for dolomite [3]. One could of course, attempt to do so via a set of assumptions regarding the chemical conditions, but the limitation is we do not know *how much dolomite is formed*, only its percentage of the total mineral mass over the run.

My other criticism is the pitch of the paper itself. Divalent ions that hydrate strongly (e.g. Fe²⁺) and for which Mg²⁺ must thus compete for solvent in concentrated solutions having reduced osmotic coefficients are indeed likely to act promote the overall process. However, what is critical concerning the question of low temperature dolomite formation is whether Zn in seawater — and solutions associated with dolomite, either as a direct precipitate or alteration product, must have some connection to seawater, as this is the source of sufficient Mg — can catalyze formation of this mineral. This is really the critical aspect in terms of long term carbon cycle relevance. The authors do state their intention to proceed in the direction of low temperature experiments, but then why cast *this* manuscript as “The role of zinc in dolomitization and the inorganic carbon cycle”? Zinc is also present in seawater at concentrations far lower than what is used here: can the authors defend a catalytic role for Zn at the nM concentrations typical of surface seawater?

Overall, I find this paper to have interesting data, and clearly the authors have expended significant experimental effort, but they do not make much complementary effort at understanding the reaction kinetics, the real “dolomite problem”. The ms. fails to cite important papers, and does not seem to understand the significance of the papers it does cite. These are serious flaws that should be corrected. In its current form, I cannot recommend publication this ms. in a flagship journal such as Nature.

Review notes —

- p. 2, L. 46 There is a critical distinction between magnesium-rich calcite (“VHMC”) and poorly ordered Ca-rich “protodolomites”: microcalorimetric data show a far greater endothermic penalty of the substitution of calcium for magnesium than of magnesium for calcium in calcite [6].
- p. 8, L. 184–186 “In this work, we have documented reaction mechanisms and rates of calcite-to-dolomite replacement in a series of saline solutions. The experiments also demonstrate that Zn ions counteract the inhibiting effect of sulfate ions on dolomite formation.” First, the term *reaction mechanism* refers to elementary reactions, i.e., those that actually occur as written. The data, although interesting, supply no real insight into actual reaction mechanism, only the overall relative abundance (solely on the basis of diffraction data) of reactants, products, intermediate or secondary phases. Data on the surface area of calcite are available (relatively high at 8.7 m²/g), although it’s not apparent why they are even included, as the surface area of the nascent dolomite or other phases itself is unknown. Second, the notion of sulfate exerting an inhibition to dolomite crystal growth is based on high temperature work of Baker and Kastner [7], and appears frequently in the literature, but has never been demonstrated conclusively at low temperatures. In contrast, microbial reduction of sulfate and the increase in bicarbonate as a result yields increases in saturation state with respect to all carbonates. The presence of sulfate and ion pair formation (MgSO₄) may indeed act as a catalyst for incorporation of Mg into carbonate [4, 5], and has been shown to have little direct negative effect on dolomite formation at low temperatures [1, 2].

REFERENCES

- [1] F. Zhang, H. Xu, H. Konishi, J. M. Kemp, E. E. Roden, and Z. Shen, “Dissolved sulfide-catalyzed precipitation of disordered dolomite: Implications for the formation mechanism of sedimentary dolomite,” *Geochimica et Cosmochimica Acta*, vol. 97, pp. 148–165, 2012. DOI: 10.1016/j.gca.2012.09.008.
- [2] M. Sánchez-Román, J. A. McKenzie, A. de Luca Rebello Wagener, M. A. Rivadeneyra, and C. Vasconcelos, “Presence of sulfate does not inhibit low-temperature dolomite precipitation,” *Earth and Planetary Science Letters*, vol. 285, pp. 131–139, 2009. DOI: 10.1016/j.epsl.2009.06.003.
- [3] R. S. Arvidson and F. T. Mackenzie, “The dolomite problem; control of precipitation kinetics by temperature and saturation state,” *American Journal of Science*, vol. 299, no. 4, pp. 257–288, 1999. DOI: 10.2475/ajs.299.4.257.
- [4] P. V. Brady, H. W. Papenguth, and J. W. Kelly, “Metal sorption to dolomite surfaces,” *Applied Geochemistry*, vol. 14, pp. 569–579, 1999. DOI: 10.1016/S0883-2927(98)00085-7.
- [5] P. V. Brady, J. L. Krumhansl, and H. W. Papenguth, “Surface complexation clues to dolomite growth,” *Geochimica et Cosmochimica Acta*, vol. 60, pp. 727–731, 1996. DOI: 10.1016/0016-7037(95)00436-X.
- [6] L. Chai, A. Navrotsky, and R. J. Reeder, “Energetics of calcium-rich dolomite,” *Geochimica et Cosmochimica Acta*, vol. 59, no. 5, pp. 939–944, 1995, ISSN: 0016-7037. DOI: 10.1016/0016-7037(95)00011-9.
- [7] P. A. Baker and M. Kastner, “Constraints on the formation of sedimentary dolomite,” *Science*, vol. 213, no. 4504, pp. 214–216, 1981. DOI: 10.1126/science.213.4504.214.

We thank the editor and the reviewers for their helpful comments. We have revised the manuscript taking account of all suggestions, and provide a detailed response here below. The reviewers' comments are in *italics* and our responses in non-italic letters. The line numbers correspond to the line numbers in the revised document with track changes.

Detailed reply to the reviewers' comments

Reviewers' comments:

Reviewer #1 (Remarks to the Author):

Very interesting paper, with some inaccuracies, though (e.g. the fact that Zn dolomite is only associated with weathering: this is not true because several hydrothermal dolomites have Zn in their lattice).

Reply:

We are very happy to read that you find the paper very interesting. Thank you also for pointing out the inaccuracy. Indeed, “weathering” does not cover all Zn dolomite, as this only relates to low temperature surface alteration. We have adapted our statement in the text to “Hydrothermal dolomite hosting MVT deposits may contain ppm to % contents of Zn, and zincian dolomite has been considered a peripheral hydrothermal hypogene alteration product in Polish ore deposits, and a low temperature supergene alteration product of primary sulphide deposits in Jabali (Yemen), Iglesias (SW Sardinia), and Yanque (Peru).”

Hydrothermal dolomite can indeed contain zinc, and there are differing interpretations on how those zincian dolomites form. Based on reviews on non-sulfide zinc ores (Boni and Mondillo, 2015; Hitzman et al., 2003) and other papers, zincian dolomite in most described deposits is interpreted to result from supergene alteration (weathering). Stable carbon and oxygen isotopes and fluid inclusion analysis have been used in interpretations on supergene versus hypogene origin of carbonate deposits. Hitzman et al. (2003) states that supergene deposits are the most common type of nonsulfide zinc deposits, and the majority of those occur in carbonate host rocks. There are only few examples of hypogene nonsulfide zinc deposits, but there are some linked to structurally controlled hydrothermal dolomitization, and it is proposed that the nonsulfide mineral assemblage (predominantly willemite and hematite) is formed by replacement of the hydrothermal dolomite.

As the reviewer comments, zincian dolomite has indeed been reported also in Tsumeb, SW Africa, for example, in a paper by Hurlbut (1957). No interpretation was made in this paper in terms of the origin; the paper does not make it clear whether limestone was replaced by zincian dolomite, or whether calcite was replaced by hydrothermal dolomite, and this dolomite then replaced by zincian dolomite, which is a model proposed in more recent publications.

In the revised manuscript, we have now stated that dolomite can contain Zn, and present the two most common interpretations on the origin of zincian dolomite (lines 123-127). We think these changes address the reviewer's concern on this front.

The experiments look exact to me, but I have still some problems in imagining a dolomitization process on an already hard limestone by this kind of fluids in such a short time as that envisaged by the AA.

Reply:

Thank you for this comment, which touches upon the translation of lab experiments and data to field scale models.

Dolomitization of hard tight limestone in deep burial conditions often happens by structurally-controlled hydrothermal dolomitization whereby faults and fractures provide a permeable pathways for dolomitizing fluids, e.g. (Vandeginste et al., 2015; Vandeginste et al., 2014; Vandeginste et al., 2013; Vandeginste et al., 2005; Vandeginste et al., 2007). So, the fluids would flow more quickly along open space in fractures (and start the dolomitization process at the rim of the fractures) than they penetrate and diffuse more deeply into the rock. This is also why hydrothermal dolomites have a fairly thin extent laterally from the fault/fracture and can extend very far along the fault/fracture (Vandeginste et al., 2014).

The goal of our paper is to demonstrate the impact of the addition of zinc ions in saline fluids on the rate of dolomitization. Translating data from laboratory experiments to what happens underground is a global challenge that applies to any similar laboratory study. In laboratory experiments, we aim at understanding the fundamental geochemistry. We use controlled conditions, so that we can extract the influence of a certain parameter and understand how this affects our system. In our case, this has been the impact of zinc. Experimental geochemistry testing parameters requires relatively fast rates to make the research feasible. The powder sample with high surface area, allows abundant contact between the fluid and the mineral, and such method is used in the majority of kinetic studies on mineral dissolution or formation. To translate our data to subsurface conditions, we need to consider the same reactive surface area, the same chemistry of the fluids, and exposure of the rock surface to fluid in similar ratio as in the lab conditions. A hard limestone is different than a powder, but we can take account of the same reactive surface area, and fluid/rock ratio. In the subsurface, tight limestone may have fractures where the rock is preferentially in contact with the fluids. In such case, dolomitization will start happening along those fractures (Vandeginste et al., 2014), as reaction with fluids takes place. On large scale, we could consider geological flow models and integrate reactive transport modelling, which could be of interest for a future subsurface case study, and is beyond the scope of this present study.

The relationships between dolomitization (the word hydrothermal was NEVER used) and MVT deposits are, in my opinion, overinterpreted and based on cursory evidence only. Of course MVT are often contained in dolomitized lithotypes, but this is not enough to link them to the same fluids, especially if the two processes are taking place at different times.

The AA mention several times that their results are important because there is a known connection between MVT Zn-Pb deposits and dolomitization, but they never discuss this connection in detail: if the dolomitizing fluid (with Zn in solution) will also precipitate Zn sulfides afterwards, or what will happen to the Zn that was catalysing the dolomitization process?

If the AA. can at least hypothesize (and describe) the relationships between dolomitization and MVT ores (and not only their spatial association), this would be a great addition to the paper. Otherwise, this aspect should not be specifically mentioned.

Reply:

The reviewer raises a very interesting point, thank you for this.

Regarding the term “hydrothermal”, we have now added this term in the manuscript. Some research suggests that the term “hydrothermal dolomite” should only be used if there is abundant evidence that the dolomite was formed at a higher temperature than the ambient temperature (Machel and Lonnee, 2002). The latter authors suggest that even a dolomite that was formed at 40°C could be called hydrothermal as long as this temperature was higher than the ambient temperature. In the setting of MVT and structurally controlled dolomite, the dolomite is considered hydrothermal, and we have now made this clear in the manuscript (lines 123, 230-238).

Regarding the concern on the genetic link between dolomite and MVT ores, we have added a concise discussion section in the paper, the first paragraph under “Influence of zinc on the carbonate formation”, as advised by the reviewer (lines 166-174). Dolomites are known to have formed at Pine Point MVT deposits both concurrently (Gleeson and Turner, 2007; Krebs and Macqueen, 1984) as well as post-ore stage (Krebs and Macqueen, 1984) with the ore sulfur isotopic composition being consistent with seawater sulfate (Sasaki and Krouse, 1969). Many MVT ore bodies are surrounded by a halo of hydrothermal dolomite (Leach et al., 2005; Paradis et al., 2007; Reid et al., 2013), which suggests that dolomitization may have occurred during the same mineralization event. Research has shown that it can be challenging to provide unequivocal evidence for simultaneous formation of sulfide ores and dolomite based on textural or geochemical measurements, and thus, the temporal and genetic relationship between host rock dolomitization and ore formation may be uncertain (Wilkinson et al., 2009). Estimates indicate that it may take 50,000 to 5,000,000 years for MVT ore deposits to form (Garven et al., 1999; Leach et al., 2001), and dolomitization and Zn-Pb sulfide formation can be overlapping stages in such mineralization event (Symons et al., 2015), which involve the same hydrothermal system (Savard et al., 2000) (Soussi et al., 2010). Sparry dolomite formation in the Ozark MVT hydrothermal system (from the Reelfoot rift) has been determined to come from the same fluid as the MVT deposit based on CL microstratigraphy and fluid inclusion temperatures and compositions (Leach et al., 1997). The two main models proposed for MVT ore formation include: 1) the mixing model, whereby metal ions and reduced sulfur are transported in two separate fluids, and then mix at the site of ore formation, and 2) the in situ reduction model, whereby metal ions and sulfate are carried in the same hydrothermal fluids and sulfate is reduced at the site of ore formation. Both models take account of the fact that high concentrations of metals cannot co-exist in a fluid together with high concentrations of reduced sulfur. These models thus suggest that ore sulfide precipitation depends on availability of reduced sulfur, and metal-rich brines are assumed to be part of the hydrothermal system in both models. Hence, it is likely that metal-rich brines are involved in the formation of dolomite during the mineralization event involving this hydrothermal system.

The majority of the zinc remains in the fluid after dolomitization. While this could lead to a progressively increasing zinc concentration in the remaining fluid upon dolomitization, the MVT ore formation is probably mainly enhanced by a higher fluid flow at the core of the MVT ore zone, compared to much lower fluid flow in the dolomitized halo.

To address these concerns of the reviewer, we have added a discussion paragraph in the revised manuscript (lines 230-238).

Reviewer #2 (Remarks to the Author):

Review of “The role of zinc in dolomitization and the inorganic carbon cycle”, by Veerle Vandeginste, Oliver Snell, Matthew R. Hall, Elisabeth Steer, and Arne Vandeginste, submitted to Nature Communications (ms# NCOMMS-18-24268-T).

Summary and recommendation —

This paper describes the results of hydrothermal experiments (200°C), that show addition of ZnCl₂ in Na-Ca-Mg-Cl brines increases the net conversion of powdered calcite to dolomite relative to Zn-free control solutions of similar ionic strength. The primary data indicate that at these temperatures, this complex multi-step reaction is accelerated relative to their controls. All well and good. However, the authors' experiments and treatment of the data derived from them supply little insight and no hard numbers of the kinetics of the process

itself. There are no data to evaluate the rate of the process in conventional units of mineral kinetics (mass per unit area per unit time), no calculations as to the in situ conditions driving this process. Thus we can not place these results in the context of published (but uncited here) rate laws for dolomite [3]. One could of course, attempt to do so via a set of assumptions regarding the chemical conditions, but the limitation is we do not know how much dolomite is formed, only its percentage of the total mineral mass over the run.

Reply:

Thank you very much for these comments. These are very good points, and we agree that the additional work, calculations and models we have carried out in terms of the kinetics of our experiments have now much improved our manuscript. We have also compared our results to existing literature. The rate law that the reviewer cites (Arvidson and Mackenzie, 1999) is indeed a well known work on dolomite. Because the cited work focuses on precipitation of dolomite, in contrast to our work on replacement of calcite to dolomite (without the addition of dissolved carbonate in solution), we cannot directly compare our results, since our method prevents calculation of the saturation index (due to lack of dissolved carbonate in solution) which is the base of the cited work. Nevertheless, we have compared our kinetic results to other published work on dolomite formation. The kinetic results are provided in the section on “Effect of zinc on dolomitization rate” (new text in lines 168-190, 193-195) and we have also added a new figure (Figure 5), which presents the results of the Avrami kinetic models of the results of our different experiments. The Avrami plots and models enabled us to calculate the rate constant and time exponent.

As mentioned, comparison with Arvidson and Mackenzie (1999) is not straightforward because we have studied calcite-to-dolomite replacement reaction, and not dolomite precipitation from a saturated fluid, and their presented model is based on the saturation index. Although direct precipitation of dolomite from a fluid does occur in nature, replacement of limestone is more common throughout the wide range of temperatures, hence, our decision on testing the calcite-to-dolomite replacement reaction in our study rather than direct dolomite precipitation reaction. The calcite-to-dolomite replacement experiments involve coupled calcite dissolution and are different to dolomite precipitation experiments. Arvidson and Mackenzie (1999) present quantification and modeling of the precipitation rate of dolomite by a rate law as a function of saturation index. Their experiments involved determining the dolomite precipitation rate in a dolomite-seeded flow reactor, and they varied temperature (100 to 200°C) and saturation index (1 to 100). Their input solutions contained besides CaCl₂ and MgCl₂, also NaHCO₃, and various CO₂ pressures. The saturation index of the solutions can be calculated from the concentrations of Ca, Mg and carbonate ions in the input solutions. (In contrast, in our experiments, we do not add dissolved carbonate in our input solutions, but rather calcite powder.) Direct comparison of our results with their experiments is challenging because of the calculation of the saturation index, which is not a controlled value in our experiments due to the dependence of the concentration of dissolved carbonate in solution on coupled calcite dissolution during the calcite-to-dolomite replacement reaction. Arvidson and Mackenzie (1999) propose the rate law $r = k (\Omega - 1)^n$ whereby the rate constant k is an Arrhenius function of temperature. Without value for saturation index, we cannot calculate the rate, but using their equation, we can calculate the rate constant for a reaction at 200 °C, as they also present a value for the activation energy of 31.9 kcal mol⁻¹ and pre-exponential factor ($10^{1.05}$), yielding a rate constant of $2 \times 10^{-14} \text{ s}^{-1}$. For our experiments and kinetic calculations, we obtain rate constants using the empirical Avrami model, which has been successfully used in kinetic studies for several mineral reactions and mineral replacement reactions (Wang et al., 2005; Xia et al., 2009). The kinetic Avrami model was also proposed for the growth of protodolomite by Rodriguez-Blanco et al. (2015). This model states that the reaction extent $y = 1 - \exp(-[k(t-t_0)]^n)$ whereby k is the

rate constant, t is the reaction time and t_0 the induction time, and n is a time exponent that depends on the reaction mechanism. A plot of $\ln [\ln (1/(1-y))]$ versus $\ln t$ should be linear if the activation energy is constant (and temperature as well) over the course of the reaction, thus if the reaction occurs under near steady state conditions and the reaction mechanism remained the same. The Avrami model is thus based on reaction extent, which takes account of mass fraction; this is different than the method used in Arvidson and MacKenzie (1999) that is common for dissolution reactions taking account of reactive surface area (and where induction time is generally not determined). Rate constants for dolomite formation reported by Rodriguez-Blanco et al. (2015) are $9 \times 10^{-10} \text{ s}^{-1}$ at $180 \text{ }^\circ\text{C}$ and $34 \times 10^{-10} \text{ s}^{-1}$ at $220 \text{ }^\circ\text{C}$. The time exponent was not reported in those papers.

Based on our own experiments, the Avrami model led us to derive a rate constant of $3.9 \times 10^{-5} \text{ s}^{-1}$ and time exponent of 2.4 for the control dolomitization reaction at $200 \text{ }^\circ\text{C}$; a rate constant of $7.6 \times 10^{-5} \text{ s}^{-1}$ and time exponent of 1.6 for the dolomitization experiments with ZnCl_2 ; and similarly, a rate constant of $7.2 \times 10^{-5} \text{ s}^{-1}$ and time exponent of 1.8 for the dolomitization experiments with ZnO . While these rate constants are orders of magnitude higher than those reported by Rodriguez-Blanco et al. (2015), the induction times in our experiments (4.5-10.5 hours) are significantly longer than those in their experiments (about 10 minutes).

Estimates on the average rate of dolomitization in the reaction (following induction time) in our experiments is calculated to be $2.3 \times 10^{-8} \text{ mol m}^{-2} \text{ s}^{-1}$ for the control series, $5.3 \times 10^{-8} \text{ mol m}^{-2} \text{ s}^{-1}$ for the experiments with ZnCl_2 and $5.0 \times 10^{-8} \text{ mol m}^{-2} \text{ s}^{-1}$ for the experiments with ZnO . These estimates assume a reactive surface area of $8.7 \text{ m}^2/\text{g}$. We have reported this here upon request by the reviewer. However, it is clear from our data that the reaction rate is not a linear function of time, and the rate changes during the reaction. Therefore, the value given is the average rate, but the Avrami equation is a better fit to the reaction kinetics in our experiments.

Regarding the concern on not knowing how much dolomite is formed, the percentage of ratio of calcite and dolomite in the reaction products can be directly correlated to reaction extent, as dolomite forms by replacement of calcite. Full conversion of 200 mg of calcite would lead to 184 mg of dolomite, taking account of molar mass of the two minerals. We checked all weights of the reaction products, including mixed calcite-dolomite products and compared the weights to expected weights by calculations that take account of the calcite/dolomite ratio in the reaction products derived from XRD analysis, and have found that the weight of all products falls within 5% error of the calculated/expected value where the calcite/dolomite ratio directly correlates with reaction extent. These results confirm the coupled calcite dissolution and dolomite precipitation reaction (and which is different to dolomite precipitation experiments with additional carbonate sources in the fluids as is the case for Arvidson and MacKenzie, 1999).

We add here below columns with the product weights for the main Zn-free and Zn-bearing experiment series. The column “total” indicates the ratio of the actual product weight to the expected product weight as calculated from the calcite/dolomite ratio, taking account of the fact that 200 mg calcite converts to 184 mg dolomite if the reaction is completed and an intermediate value will be obtained where the reaction is incomplete. These data provide evidence that the calcite and dolomite percentages directly correlate with reaction extent and that the dolomitization reaction occurs by coupled calcite dissolution and dolomite precipitation since a total product weight of 184 to 200 mg is obtained (within error). Small errors within 5% may be introduced by minor amounts of halite or other trace products as described in the text. We have added this in the revised manuscript (lines 389-393).

Zinc-free solution experiments

Time (hours)	Dolomite percentage			Product weight (mg)		
	Average	Number of analyses	Stdev	Average	Stdev	Total
12	3	4	2	194	3	0.97
13	7	3	4	193	10	0.97
14	15	3	2	200	29	1.01
15	28	3	6	198	4	1.01
16	49	4	3	193	16	1.00
17	44	3	1	189	1	0.98
18	54	3	2	200	21	1.05
19	98	3	3	188	2	1.02
20	91	4	7	189	19	1.02
21	80	3	2	190	1	1.01
22	92	3	3	189	6	1.02
23	99	3	1	188	9	1.02
24	100	4	0	182	1	0.99
25	100	3	0	179	6	0.97
26	100	3	0	178	21	0.96
27	100	3	0	192	2	1.04
32	100	2	0	192	1	1.04
36	100	1		194		1.05
40	100	1		191		1.04
44	100	1		185		1.01
48	100	1		186		1.01
52	100	1		181		0.98
56	100	1		188		1.02

Zinc-bearing (0.03M ZnCl₂) solution experiments

Time (hours)	Dolomite percentage			Product weight (mg)		
	Average	Number of analyses	Stdev	Average	Stdev	Total
1	0	1		190		0.95
2	0	1		191		0.93
3	0	1		202		1.01
4	0	3	0	189	5	0.95
5	5	3	2	191	11	0.96
6	16	3	4	197	16	1.00
7	42	3	5	184	19	0.95
8	61	5	4	193	10	1.01
9	70	3	9	192	11	1.02

10	79	3	5	195	4	1.04
11	92	3	6	179	7	0.96
12	98	3	3	186	7	1.01
13	99	3	2	185	11	1.00
14	100	3	2	179	14	0.97
15	99	3	1	183	17	0.99
16	99	3	2	183	11	1.00
20	100	1		192		1.04
24	97	1		179		0.97
32	100	1		188		1.02

My other criticism is the pitch of the paper itself. Divalent ions that hydrate strongly (e.g. Fe²⁺) and for which Mg²⁺ must thus compete for solvent in concentrated solutions having reduced osmotic coefficients are indeed likely to act promote the overall process. However, what is critical concerning the question of low temperature dolomite formation is whether Zn in seawater — and solutions associated with dolomite, either as a direct precipitate or alteration product, must have some connection to seawater, as this is the source of sufficient Mg — can catalyze formation of this mineral. This is really the critical aspect in terms of long term carbon cycle relevance. The authors do state their intention to proceed in the direction of low temperature experiments, but then why cast this manuscript as “The role of zinc in dolomitization and the inorganic carbon cycle”? Zinc is also present in seawater at concentrations far lower than what is used here: can the authors defend a catalytic role for Zn at the nM concentrations typical of surface seawater?

Reply:

Thank you for this feedback on the pitch of the paper.

Reviewer correctly points out that most fluids (apart from rainwater) are derived from seawater. Fluids in hypersaline lakes and lagoons, where dolomite is interpreted to form in near-surface conditions, are derived from seawater upon significant evaporation (McCaffrey et al., 1987). Similarly, subsurface fluids are also mainly derived from evaporation of seawater, and can also come from dissolution of evaporite layers (Hanor, 1994). Structurally controlled hydrothermal dolomite is generally formed from brines that originate from the evaporation of seawater (Vandeginste et al., 2005). Thus, seawater or brine derived from seawater is involved in the formation of dolomite both at low and high temperature. Dolomite does not seem to form in current-day seawater composition; however, it has been shown to form in hypersaline lakes with microbial activity (Vasconcelos et al., 1995). Using current-day seawater composition in the experiments would be unlikely to lead to dolomitization based on natural observations. However, modification of the fluids by evaporation, as is the case for hypersaline lakes, leads to higher concentrations of the components in the fluids. In the case of subsurface brines, in addition to more highly saline fluids than seawater due to evaporation, the fluids will also undergo reactions with the subsurface formations, which will change the composition (Hanor, 1994). In the context of dolomitization linked with MVT deposits, it is known that the fluids (which are originally derived from the evaporation of seawater) have leached metals from subsurface formations and concentrated those in the fluids. In both natural environments, it is expected that the zinc concentrations in the fluids that are at play in dolomitization are higher than the concentration in current seawater. We have discussed both the low temperature environment of peritidal and lagoonal environments and the high temperature environment of MVT settings, where dolomite formation occurs.

We do not know what the zinc concentrations were in seawater-derived fluids that were involved in dolomitization through geological time. It is possible that zinc concentrations were high in those environments. At this stage, we cannot just assume that zinc concentration in current-day seawater was exactly the same as in the past, and in particular, we do not know the zinc concentrations in past peritidal and lagoonal environments which may have been sites of massive dolomitization.

Seawater plays a big role in the inorganic carbon cycle, and is a major reservoir for carbon, and so is the carbonate reservoir of the Earth's crust. The global carbon cycle is not restricted to seawater; carbon gets trapped in sediments which then become part of the burial environment, and the formation of carbonate is not limited to the open sea environment. Fluid-rock interaction in the burial environment impacts on the cycling of elements. We have documented both low and high temperature environments in which dolomite can be formed. We have added a section in our discussion in the revised manuscript to clarify this (lines 230-238), as well as a line in the introduction (lines 42-43). We have also put less focus on the geochemical cycles in the introduction (lines 35-36 and 70-71). To further address the concerns of the reviewer on the focus on the global carbon cycle, we have now also removed "and the inorganic carbon cycle" from the title of our paper, and have also changed the last discussion section in our paper to "Influence of zinc on carbonate formation" to remove the reference to the global carbon cycle and made some changes in this discussion (lines 229-309).

Overall, I find this paper to have interesting data, and clearly the authors have expended significant experimental effort, but they do not make much complementary effort at understanding the reaction kinetics, the real "dolomite problem". The ms. Fails to cite important papers, and does not seem to understand the significance of the papers it does cite. These are serious flaws that should be corrected. In its current form, I cannot recommend publication this ms. in a flagship journal such as Nature.

Reply:

We are very happy to read that you find the paper to have interesting data. Thank you very much also for pointing out the additional work that was needed on the kinetic analysis, which we have now added to the paper (lines 168-190, 193-195), and the clarification that was needed on the actual reaction product masses (instead of only the calcite/dolomite ratios) (lines 386-393). This has much improved our manuscript and we very much appreciate your suggestions.

There are indeed many important papers on the topic of the "dolomite problem", but we have a cap on the number of publications we can cite for this contribution. We have added several papers including the ones suggested by the reviewer, and if the editor allowed us to cite more papers, we would be very keen to do so; there are so many, very good papers on this interesting topic. The dolomite problem has been well defined by many authors. One of the most comprehensive explanations of the dolomite problem may be that stated by Machel (2004), which builds on 4 aspects: 1) dolomite occurs in many different sedimentary and diagenetic settings, 2) for many case studies various genetic interpretations can be provided for dolomite formation by the data available, 3) dolomite is scarce in Holocene environments, but abundant in ancient rocks, 4) well-ordered, stoichiometric dolomite has never been successfully grown (inorganically) in laboratory experiments at near-surface conditions. In addition, Machel (2004) states that the fourth aspect necessitates that geochemical parameters needed for back-calculating the compositions of the dolomitizing fluids have to be extrapolated from high-temperature experiments. This final point further confirms the need to investigate geochemical effects at elevated temperature, since the reaction rates are too slow at low temperature to make laboratory experiments feasible. Land (1998) ran a dolomite

precipitation experiment for 32 years without success. The impact of a geochemical parameter on the formation of well-ordered stoichiometric dolomite is thus assessed at elevated temperature to make laboratory experiments feasible.

Review notes —

p. 2, L. 46 There is a critical distinction between magnesium-rich calcite (“VHMC”) and poorly ordered Ca-rich “protodolomites”: microcalorimetric data show a far greater endothermic penalty of the substitution of calcium for magnesium than of magnesium for calcium in calcite [6].

Reply:

Thank you for this comment. Dolomite has $R\bar{3}$ symmetry and calcite $R\bar{3}c$ symmetry. The dolomite symmetry is lower mainly due to the Ca-Mg ordering. High magnesium calcite with near-dolomite stoichiometry (about 50 mol% $MgCO_3$) has $R\bar{3}c$ symmetry and is referred to as protodolomite or very high magnesium calcite, which is what we referred to in our manuscript (not the higher calcium, magnesium-rich calcite). The terminology we used follows that presented in the publication by Gregg et al. (2015) in *Sedimentology*. Nevertheless, we agree that this may lead to confusion; hence, to avoid confusion and address the reviewer’s concern, we have replaced the term “very high magnesium calcite (VHMC)” by “protodolomite” in the revised manuscript (adapted throughout the manuscript).

p. 8, L. 184–186 “In this work, we have documented reaction mechanisms and rates of calcite-to-dolomite replacement in a series of saline solutions. The experiments also demonstrate that Zn ions counteract the inhibiting effect of sulfate ions on dolomite formation.” First, the term reaction mechanism refers to elementary reactions, i.e., those that actually occur as written. The data, although interesting, supply no real insight into actual reaction mechanism, only the overall relative abundance (solely on the basis of diffraction data) of reactants, products, intermediate or secondary phases. Data on the surface area of calcite are available (relatively high at 8.7 m²/g), although it’s not apparent why they are even included, as the surface area of the nascent dolomite or other phases itself is unknown. Second, the notion of sulfate exerting an inhibition to dolomite crystal growth is based on high temperature work of Baker and Kastner [7], and appears frequently in the literature, but has never been demonstrated conclusively at low temperatures. In contrast, microbial reduction of sulfate and the increase in bicarbonate as a result yields increases in saturation state with respect to all carbonates. The presence of sulfate and ion pair formation (MgSO₄) may indeed act as a catalyst for incorporation of Mg into carbonate [4, 5], and has been shown to have little direct negative effect on dolomite formation at low temperatures [1, 2].

Reply:

Thank you for having carefully read our manuscript, assessed the data and interpretations, and providing those comments.

With regards to the reaction mechanism, as shown in the tables above and mentioned in our reply also above, the weights correspond directly to the reaction of calcite-to-dolomite replacement in a stoichiometric mole-per-mole replacement reaction. Indeed, this was not clear in the first version, and we think that with the additional weight data added, this addresses the concern of the reviewer. In the revised version, we have added this explanation in the Materials and methods section and the weights of the reaction products provide direct evidence for the mole-per-mole replacement reaction of calcite to dolomite in our experiments. To avoid any further concerns on terminology of reaction mechanisms, we have replaced the term “reaction mechanism” by “reaction pathway” in the revised manuscript (adapted throughout the manuscript). We have addressed the calcite-to-dolomite replacement reaction and documented calcite dissolution coupled with the formation of protodolomite and

then transformation to well-ordered dolomite. The reaction product weights confirm that reaction product (calcite and dolomite) ratios directly correlate with the reaction extent. Regarding the use of the reactive surface area, we have added a calculation using this in the revised version of the manuscript. The surface area of the reactant can be used in the calculation of reaction rates. For example, it is commonly used in kinetics of dissolution reactions (Cubillas et al., 2005; Gudbrandsson et al., 2014; Pokrovsky et al., 2005; Stockmann et al., 2011), but also for precipitation and replacement reactions (Xia et al., 2009). In the Avrami model, the reaction rate changes during the progress of the reaction. Still, we have calculated the reaction rate in the midst of the reaction progress, and have taken account of the calcite surface area to do so.

With respect to the second point, on the effect of sulfate, there has indeed been some discussion around this. As the reviewer rightly points out, the Baker and Kastner study based on experiments at high temperature is frequently cited. Many studies on dolomite formation have indeed been conducted at elevated temperature because the more rapid kinetics at higher temperature enable testing parameters at short time scales that are readily feasible in laboratory experiments. We have included tests of the impact of sulfate in our experiments to support and confirm the counteracting effect of zinc, but the impact of sulfate was not the main goal of our study. Still, the inhibiting effect of sulfate was documented for dolomite formation at temperatures similar to those that we have used in our experiments, thus making it relevant to our experiments. The impact of sulfate has indeed not been conclusively demonstrated for low temperature conditions, and we present both opposing views in the introduction in our manuscript. Some authors argue that it is the reduced form of sulfate (i.e. sulphide) rather than sulfate itself which may catalyse dolomite formation, which is in line with the process of microbial reduction of sulfate, as mentioned by the reviewer. We have clarified these discussions and debate in the manuscripts (lines 49-54). To further address the reviewer's concern, we have made it clear in the revised manuscript that our observations for zinc and sulfate from the experiments are valid at the temperature tested in the experiments (lines 314-316).

References

- [1] F. Zhang, H. Xu, H. Konishi, J. M. Kemp, E. E. Roden, and Z. Shen, "Dissolved sulfide-catalyzed precipitation of disordered dolomite: Implications for the formation mechanism of sedimentary dolomite," *Geochimica et Cosmochimica Acta*, vol. 97, pp. 148–165, 2012. doi: 10.1016/j.gca.2012.09.008.
- [2] M. Sanchez-Roman, J. A. McKenzie, A. de Luca Rebello Wagener, M. A. Rivadeneyra, and C. Vasconcelos, "Presence of sulfate does not inhibit low-temperature dolomite precipitation," *Earth and Planetary Science Letters*, vol. 285, pp. 131–139, 2009. doi: 10.1016/j.epsl.2009.06.003.
- [3] R. S. Arvidson and F. T. Mackenzie, "The dolomite problem; control of precipitation kinetics by temperature and saturation state," *American Journal of Science*, vol. 299, no. 4, pp. 257–288, 1999. doi: 10.2475/ajs.299.4.257.
- [4] P. V. Brady, H. W. Papenguth, and J. W. Kelly, "Metal sorption to dolomite surfaces," *Applied Geochemistry*, vol. 14, pp. 569–579, 1999. doi: 10.1016/S0883-2927(98)00085-7.
- [5] P. V. Brady, J. L. Krumhansl, and H. W. Papenguth, "Surface complexation clues to dolomite growth," *Geochimica et Cosmochimica Acta*, vol. 60, pp. 727–731, 1996. doi: 10.1016/0016-7037(95)00436-X.
- [6] L. Chai, A. Navrotsky, and R. J. Reeder, "Energetics of calcium-rich dolomite," *Geochimica et Cosmochimica Acta*, vol. 59, no. 5, pp. 939–944, 1995, issn: 0016-7037. doi: 10.1016/0016-7037(95)00011-9.

[7] P. A. Baker and M. Kastner, "Constraints on the formation of sedimentary dolomite," *Science*, vol. 213, no. 4504, pp. 214–216, 1981. doi: 10.1126/science.213.4504.214.

Reply:

Thank you for this. We are familiar with those papers, and cited several of those in our original manuscript. We have now also added other papers from this list. I think the editor will decide whether we are allowed to go beyond the cap on number of citations, in order to address the reviewer's concerns.

References

- Arvidson, R.S., Mackenzie, F.T., 1999. The dolomite problem: Control of precipitation kinetics by temperature and saturation state. *American Journal of Science*, 299(4): 257-288.
- Boni, M., Mondillo, N., 2015. The "Calamines" and the "Others": The great family of supergene nonsulfide zinc ores. *Ore Geology Reviews*, 67: 208-233.
- Cubillas, P., Kohler, S., Prieto, M., Chairat, C., Oelkers, E.H., 2005. Experimental determination of the dissolution rates of calcite, aragonite, and bivalves. *Chemical Geology*, 216(1-2): 59-77.
- Garven, G., Appold, M.S., Toptygina, V.I., Hazlett, T.J., 1999. Hydrogeologic modeling of the genesis of carbonate-hosted lead-zinc ores. *Hydrogeology Journal*, 7(1): 108-126.
- Gleeson, S.A., Turner, W.A., 2007. Fluid inclusion constraints on the origin of the brines responsible for Pb-Zn mineralization at Pine Point and coarse non-saddle and saddle dolomite formation in southern Northwest Territories. *Geofluids*, 7(1): 51-68.
- Gregg, J.M., Bish, D.L., Kaczmarek, S.E., Machel, H.G., 2015. Mineralogy, nucleation and growth of dolomite in the laboratory and sedimentary environment: A review. *Sedimentology*, 62(6): 1749-1769.
- Gudbrandsson, S., Wolff-Boenisch, D., Gislason, S.R., Oelkers, E.H., 2014. Experimental determination of plagioclase dissolution rates as a function of its composition and pH at 22 degrees C. *Geochimica Et Cosmochimica Acta*, 139: 154-172.
- Hanor, J.S., 1994. Origin of saline fluids in sedimentary basins. In: Parnell, J. (Ed.), *Geofluids: Origin, Migration and Evolution of Fluids in Sedimentary Basins*. Geological Society Special Publications, pp. 151-174.
- Hitzman, M.W., Reynolds, N.A., Sangster, D.F., Allen, C.R., Carman, C.E., 2003. Classification, genesis, and exploration guides for nonsulfide zinc deposits. *Economic Geology*, 98: 685-714.
- Hurlbut, C.S., 1957. ZINCIAN AND PLUMBIAN DOLOMITE FROM TSUMEB, SOUTHWEST AFRICA. *American Mineralogist*, 42(11-2): 798-803.
- Krebs, W., Macqueen, R., 1984. Sequence of diagenetic and mineralization events: Pine Point lead-zinc property, NWT, Canada. *Bulletin of Canadian Petroleum Geology*, 32: 434-464.
- Land, L.S., 1998. Failure to precipitate dolomite at 25°C from dilute solution despite 1000-fold oversaturation after 32 years. *Aquatic Geochemistry*, 4: 361-368.
- Leach, D.L. et al., 2001. Mississippi Valley-type lead-zinc deposits through geological time: implications from recent age-dating research. *Mineralium Deposita*, 36(8): 711-740.
- Leach, D.L. et al., 2005. Sediment-hosted lead-zinc deposits: a global perspective. *Economic Geology*, 100: 561-607.
- Machel, H.G., 2004. Concepts and models of dolomitization: a critical reappraisal. In: Braithwaite, C.J.R., Rizzi, G., Darke, G. (Eds.), *The geometry and petrogenesis of*

- dolomite hydrocarbon reservoirs. Geological Society, London, Special Publications. The Geological Society of London, London, pp. 7-63.
- Machel, H.G., Lonnee, J., 2002. Hydrothermal dolomite - a product of poor definition and imagination. *Sedimentary Geology*, 152(3-4): 163-171.
- McCaffrey, M.A., Lazar, B., Holland, H.D., 1987. THE EVAPORATION PATH OF SEAWATER AND THE COPRECIPITATION OF BR- AND K+ WITH HALITE. *Journal of Sedimentary Petrology*, 57(5): 928-937.
- Paradis, S., Hannigan, P., Dewing, K., 2007. Mississippi Valley-type lead-zinc deposits. In: Goodfellow, W.D. (Ed.), *Mineral deposits of Canada: A synthesis of major deposit-types, strict metallogeny, the evolution of geological provinces, and exploration methods*. Geological Association of Canada, Mineral Deposits Division, Special Publication, pp. 185-203.
- Pokrovsky, O.S., Golubev, S.V., Schott, J., 2005. Dissolution kinetics of calcite, dolomite and magnesite at 25 degrees C and 0 to 50 atm pCO(2). *Chemical Geology*, 217(3-4): 239-255.
- Reid, S., Dewing, K., Sharp, R., 2013. Polaris as a guide to northern exploration: Ore textures, paragenesis and the origin of the carbonate-hosted Polaris Zn-Pb Mine, Nunavut, Canada. *Ore Geology Reviews*, 51: 27-42.
- Rodriguez-Blanco, J.D., Shaw, S., Benning, L.G., 2015. A route for the direct crystallization of dolomite. *American Mineralogist*, 100: 1172-1181.
- Sasaki, A., Krouse, H.R., 1969. SULFUR ISOTOPES AND PINE POINT LEAD-ZINC MINERALIZATION. *Economic Geology*, 64(7): 718-&.
- Savard, M.M., Chi, G., Sami, T., Williams-Jones, A.E., Leigh, K., 2000. Fluid inclusion and carbon, oxygen, and strontium isotope study of the Polaris Mississippi Valley-type Zn-Pb deposit, Canadian Arctic Archipelago: implications for ore genesis. *Mineralium Deposita*, 35(6): 495-510.
- Stockmann, G.J., Wolff-Boenisch, D., Gislason, S.R., Oelkers, E.H., 2011. Do carbonate precipitates affect dissolution kinetics? 1: Basaltic glass. *Chemical Geology*, 284(3-4): 306-316.
- Symons, D.T.A., Tornos, F., Kawasaki, K., Velasco, F., Rosales, I., 2015. Genetic constraints from paleomagnetic dating for the Aliva zinc-lead deposit, Picos de Europa Unit, northern Spain. *Mineralium Deposita*, 50: 953-966.
- Vandeginste, V., John, C.M., Beckert, J., 2015. Diagenetic Geobodies: Fracture-Controlled Burial Dolomite in Outcrops From Northern Oman. *SPE Reservoir Evaluation & Engineering*, 18(1): 84-93.
- Vandeginste, V., John, C.M., Cosgrove, J.W., Manning, C., 2014. Dimensions, texture-distribution and geochemical heterogeneities of fracture-related dolomite geobodies hosted in Ediacaran limestone, northern Oman. *AAPG Bulletin*, 98(9): 1789-1809.
- Vandeginste, V., John, C.M., van de Flierdt, T., Cosgrove, J.W., 2013. Linking process, dimension, texture and geochemistry in dolomite geobodies: a case study from Wadi Mistal (northern Oman). *AAPG Bulletin*, 97(7): 1181-1207.
- Vandeginste, V. et al., 2005. Zebra dolomitization as a result of focused fluid flow in the Rocky Mountains Fold and Thrust Belt, Canada. *Sedimentology*, 52: 1067-1095.
- Vandeginste, V. et al., 2007. Geochemical constraints on the origin of the Kicking Horse and Monarch Mississippi Valley-type lead-zinc ore deposits, southeast British Columbia, Canada. *Mineralium Deposita*, 42(8): 913-935.
- Vasconcelos, C., McKenzie, J.A., Bernasconi, S., Grujic, D., Tien, A.J., 1995. Microbial mediation as a possible mechanism for natural dolomite formation at low temperatures. *Nature*, 377(6546): 220-222.

- Wang, H.P., Pring, A., Ngothal, Y., O'Neill, B., 2005. A low-temperature kinetic study of the exsolution of pentlandite from the monosulfide solid solution using a refined Avrami method. *Geochimica Et Cosmochimica Acta*, 69(2): 415-425.
- Wilkinson, J.J., Stoffell, B., Wilkinson, C.C., Jeffries, T.E., Appold, M.S., 2009. Anomalously Metal-Rich Fluids Form Hydrothermal Ore Deposits. *Science*, 323(5915): 764-767.
- Xia, F. et al., 2009. Mechanism and kinetics of pseudomorphic mineral replacement reactions: A case study of the replacement of pentlandite by violarite. *Geochimica Et Cosmochimica Acta*, 73(7): 1945-1969.

Reviewers' comments:

Reviewer #1 (Remarks to the Author):

I think that the authors have addressed satisfactorily the questions of this reviewer. However, I would stress AGAIN and MORE that this is a laboratory experiment, and that the association between Zn-associated dolomite and MVT ores is not yet proven, unless (as the authors say) both phenomena can occur spatially together. Of course (but then the story would be going much further) there is the problem of carbonate-hosted SEDEX as HVC and others...

Review of “The role of zinc in dolomitization and the inorganic carbon cycle”, by Veerle Vandeginste, Oliver Snell, Matthew R. Hall, Elisabeth Steer, and Arne Vandeginste, (revision) submitted to *Nature Communications* (ms# NCOMMS-18-24268-T).

Summary and recommendation — In my initial review of this paper, my primary concerns were

- 1) the absence of any real treatment of the kinetics of dolomite formation — whether by dissolution-replacement reaction (via growth of dolomite driving coupled dissolution of calcite, or complex *in situ* conversion, in which the distinction involves volume change, void space, etc.), or primary formation — even to the point of understanding how much dolomite formed over the course of the experiment;
- 2) a problematic cast of the relevance of their results to the carbon cycle, given Zn availability in seawater – again, this relates to what natural environment the authors view their results relevant; and
- 3) a haphazard treatment of relevant literature.

Aside from relatively straightforward changes (replacement of the original references to “VHMC” with “protodolomite”, rewording in references to the carbon system, etc.), the most important aspect of the authors’ revision is the fitting of their conversion data using the Avrami [4] (Johnson-Mehl-Avrami-Kohnogorov) equation. This approach has been used previously, most notably by Sibley and coworkers (e.g. [2, 3]). Just to review, Avrami-type treatments are often applied to transformation reactions occurring entirely within the solid state, focusing on the time-dependence of extent of conversion of a parent to a product phase via nucleation and growth. In the simplest form of common equation, the extent of conversion x is cast as an exponential function of time:

$$x(t) = \exp[-(t/\tau)^m]$$

where τ contains terms for nuclei production rate and growth rate (isotropic velocity of the migrating reaction front), often assumed to be constant. The interactions of growing nuclei are assumed to be limited only by mutual impingement. The value of m , the so-called Avrami exponent expresses the dimensionality: $m = 3$ for 3D spheres, $m = 2$ for 2D disks, $m = 1$ for 1D rods; larger or fractional values appear for irregular, complex morphologies. For fitting purposes, the equation can be linearized (see Sharp-Hancock [1]) to yield

$$\ln[-\ln[1-x(t)]] = m \ln t + \beta$$

allowing simple regression of parameters (m, β, \dots) with small error within the range $0.15 \leq x \leq 0.5$ [1]. These equations are widely used in solid phase reactions because of the simple representation of the nucleation and growth rate of the new phase at the expense of the old, emphasizing that it is this substrate that is reacting: the role of the dissolved phase (if even present) is ignored. In the case of the nucleation of dolomite at the expense of pure calcite, it is thus assumed that access to magnesium is also not limiting (as these expressions provide no means of even representing concentration dependencies).

My major comment here is that, although this approach certainly documents the extent and rate of conversion of (in this case) powered calcite to dolomite —with and without Zn, the approach is unlikely to illuminate the *role* of Zn (the major point of the paper). Avrami parameters simply allow easy fits to conversion data, but are unlikely to supply much insight into *how* Zn participates in the surface-controlled reaction mechanisms. For example, I would be careful about placing much emphasis on the duration of *induction periods*. Experimentally, an induction period is simply the time one must wait to *detect* results – thus, it is a limitation of the experimental method (and may or may not have anything relevant to say about the process itself). To be clearer, exactly how would one define when the clock in a natural setting “starts”? Is this a purely stochastic process, when calcite is first brought into contact with fluids? If the role of Zn is to enhance Mg availability (homogeneously within solution), this process is essentially instantaneous. My point is that introducing an induction period as some kind of magic waiting period again sidesteps the issue of what is really happening with the fluid-mineral system.

The authors have a nice paper, it certainly raises the possibility that Zn can catalyze dolomitization reactions, and have addressed most of my initial complaints in this revision. However, as they also express in their conclusions, the application of these results to the low or moderate temperatures of shallow crustal environments needs to explicitly document the role that Zn plays. I include only one other minor comment below. Otherwise I recommend acceptance of this revision.

Review notes —

- (1) Line 159. The authors refer to “high supersaturation levels that are reached when carbonate is in solution”. But there are no data (unless I missed them) for the extent of supersaturation with respect to the nascent protodolomite. If some fraction of the calcite dissolves to saturation, the authors could at least use that constraint together with their initial bulk concentrations to estimate a CO_3^{2-} ion activity and a ΔG , recognizing that the solubility of the initial phase may be quite high (due both to disorder as well as critical size effects). I’m not insisting that the authors supply this information – but I would remove such inferences, as I think it detracts from their results to infer conclusions about saturation state without at least the requisite calculation in hand.

Rolf S. Arvidson
University of Bremen

REFERENCES

- [1] J. D. Hancock and J. H. Sharp, “Method of comparing solid-state kinetic data and its application to the decomposition of kaolinite, brucite, and BaCO_3 ,” *Journal of the American Ceramic Society*, vol. 55, no. 2, pp. 74–77, DOI: 10.1111/j.1151-2916.1972.tb11213.x.

- [2] D. F. Sibley, "Unstable to stable transformations during dolomitization," *The Journal of Geology*, vol. 98, no. 5, pp. 739–748, 1990. DOI: 10.1086/629437.
- [3] D. F. Sibley, R. E. Dedoes, and T. R. Bartlett, "Kinetics of dolomitization," *Geology*, vol. 15, no. 12, pp. 1112–1114, 1987. DOI: 10.1130/0091-7613(1987)15<1112:KOD>2.0.CO;2.
- [4] M. Avrami, "Kinetics of phase change. I General theory," *The Journal of Chemical Physics*, vol. 7, no. 12, pp. 1103–1112, 1939. DOI: 10.1063/1.1750380.

We thank the editor and the reviewers for their helpful comments. We have revised the manuscript taking account of all suggestions, and provide a detailed response here below. The reviewers' comments are in *italics* and our responses in non-italic letters. The line numbers correspond to the line numbers in the revised document with track changes.

Detailed reply to the reviewers' comments

Reviewers' comments:

Reviewer #1 (Remarks to the Author):

I think that the authors have addressed satisfactorily the questions of this reviewer. However, I would stress AGAIN and MORE that this is a laboratory experiment, and that the association between Zn-associated dolomite and MVT ores is not yet proven, unless (as the authors say) both phenomena can occur spatially together. Of course (but then the story would be going much further) there is the problem of carbonate-hosted SEDEX as HYC and others...

Reply:

Thank you for approving the changes based on previous comments.

We agree with the reviewer. Our work presents indeed laboratory experiments, testing whether the presence of dissolved zinc in saline solutions impacts the rate of calcite-to-dolomite replacement. One of our drives to test the impact of dissolved zinc was based on dolomite and Mississippi Valley-type (MVT) ores occurring spatially together, but we do not present results of field work here, and previous field studies on MVT ores and dolomite still debate whether there may be a genetic link between the two where they occur spatially together.

To address this concern, we have reworded part of the abstract to make sure we only refer to a not uncommon spatial association, but no genetic link. "... we demonstrate an unexpected acceleration of dolomite formation in zinc-enriched saline fluids, reflecting a not uncommon spatial association of dolomite with Mississippi Valley-type Zn ores" (line 19-20).

Similarly, we have reworded this in the introduction "... and in burial settings where dolomite may spatially occur alongside ore deposits." (line 47)

We have also slightly reworded the text in lines 204-205 "... may have caused ..."

Reviewer #2 (Remarks to the Author):

Review of "The role of zinc in dolomitization and the inorganic carbon cycle", by Veerle Vandeginste, Oliver Snell, Matthew R. Hall, Elisabeth Steer, and Arne Vandeginste, (revision) submitted to Nature Communications (ms# NCOMMS-18-24268-T).

Summary and recommendation — In my initial review of this paper, my primary concerns were

1) the absence of any real treatment of the kinetics of dolomite formation — whether by dissolution-replacement reaction (via growth of dolomite driving coupled dissolution of calcite, or complex in situ conversion, in which the distinction involves volume change, void space, etc.), or primary formation — even to the point of understanding how much dolomite formed over the course of the experiment;

2) a problematic cast of the relevance of their relevance to the carbon cycle, given Zn availability in seawater – again, this relates to what natural environment the authors view their results relevant; and

3) a haphazard treatment of relevant literature.

Aside from relatively straightforward changes (replacement of the original references to “VHMC” with “protodolomite”, rewording in references to the carbon system, etc.), the most important aspect of the authors’ revision is the fitting of their conversion data using the Avrami [4] (Johnson-Mehl-Avrami-Kohnogorov) equation. This approach has been used previously, most notably by Sibley and coworkers (e.g. [2, 3]). Just to review, Avrami-type treatments are often applied to transformation reactions occurring entirely within the solid state, focusing on the time-dependence of extent of conversion of a parent to a product phase via nucleation and growth. In the simplest form of common equation, the extent of conversion x is cast as an exponential function of time:

$$x(t) = \exp[-(t/\tau)^m]$$

where τ contains terms for nuclei production rate and growth rate (isotropic velocity of the migrating reaction front), often assumed to be constant. The interactions of growing nuclei are assumed to be limited only by mutual impingement. The value of m , the so-called Avrami exponent expresses the dimensionality: $m = 3$ for 3D spheres, $m = 2$ for 2D disks, $m = 1$ for 1D rods; larger or fractional values appear for irregular, complex morphologies. For fitting purposes, the equation can be linearized (see Sharp-Hancock [1]) to yield

$$\ln[-\ln [1 - x(t)]] = m \ln t + \beta$$

allowing simple regression of parameters (m, β, \dots) with small error within the range $0.15 \leq x \leq 0.5$ [1]. These equations are widely used in solid phase reactions because of the simple representation of the nucleation and growth rate of the new phase at the expense of the old, emphasizing that it is this substrate that is reacting: the role of the dissolved phase (if even present) is ignored. In the case of the nucleation of dolomite at the expense of pure calcite, it is thus assumed that access to magnesium is also not limiting (as these expressions provide no means of even representing concentration dependencies).

My major comment here is that, although this approach certainly documents the extent and rate of conversion of (in this case) powdered calcite to dolomite—with and without Zn, the approach is unlikely to illuminate the role of Zn (the major point of the paper). Avrami parameters simply allow easy fits to conversion data, but are unlikely to supply much insight into how Zn participates in the surface-controlled reaction mechanisms. For example, I would be careful about placing much emphasis on the duration of induction periods.

Experimentally, an induction period is simply the time one must wait to detect results – thus, it is a limitation of the experimental method (and may or may not have anything relevant to say about the process itself). To be clearer, exactly how would one define when the clock in a natural setting “starts”? Is this a purely stochastic process, when calcite is first brought into contact with fluids? If the role of Zn is to enhance Mg availability (homogeneously within solution), this process is essentially instantaneous. My point is that introducing an induction period as some kind of magic waiting period again sidesteps the issue of what is really happening with the fluid-mineral system.

The authors have a nice paper, it certainly raises the possibility that Zn can catalyze dolomitization reactions, and have addressed most of my initial complaints in this revision. However, as they also express in their conclusions, the application of these results to the low or moderate temperatures of shallow crustal environments needs to explicitly document the role that Zn plays. I include only one other minor comment below. Otherwise I recommend acceptance of this revision.

Review notes —

(1) Line 159. The authors refer to “high supersaturation levels that are reached when carbonate is in solution”. But there are no data (unless I missed them) for the extent of supersaturation with respect to the nascent protodolomite.

If some fraction of the calcite dissolves to saturation, the authors could at least use that constraint together with their initial bulk concentrations to estimate a CO_3^{2-} ion activity and a ΔG , recognizing that the solubility of the initial phase may be quite high (due both to disorder as well as critical size effects). I’m not insisting that the authors supply this

information – but I would remove such inferences, as I think it detracts from their results to infer conclusions about saturation state without at least the requisite calculation in hand.

Rolf S. Arvidson

University of Bremen

References

[1] J. D. Hancock and J. H. Sharp, “Method of comparing solid-state kinetic data and its application to the decomposition of kaolinite, brucite, and BaCO₃,” *Journal of the American Ceramic Society*, vol. 55, no. 2, pp. 74–77, doi: 10.1111/j.1151-2916.1972.tb11213.x.

[2] D. F. Sibley, “Unstable to stable transformations during dolomitization,” *The Journal of Geology*, vol. 98, no. 5, pp. 739–748, 1990. doi: 10.1086/629437.

[3] D. F. Sibley, R. E. Dedoes, and T. R. Bartlett, “Kinetics of dolomitization,” *Geology*, vol. 15, no. 12, pp. 1112–1114, 1987.

doi: 10.1130/0091-7613(1987)15<1112:KOD>2.0.CO;2.

[4] M. Avrami, “Kinetics of phase change. I General theory,” *The Journal of Chemical Physics*, vol. 7, no. 12, pp. 1103–1112, 1939. doi: 10.1063/1.1750380.

Reply:

We thank Professor Arvidson for these detailed, insightful and constructive comments. Thank you for acknowledging the changes we made based on your previous suggestions; we appreciate that you find the manuscript is now improved.

We agree with the reviewer that the Avrami model is mainly used for solid phase transformations, as was previously proposed for the calcite-to-dolomite replacement; in our manuscript we had referred to two relatively recent papers that report this, and have now also added the citations to the papers by Sibley, as mentioned by the reviewer (see line 141).

In terms of the duration of the induction periods, we agree with the reviewer that there are experimental factors that contribute to the duration of the induction period. In crystal nucleation kinetics, the *true induction period* is the time duration between the onset of supersaturation in a solution and the formation of critical nuclei, and this period depends on solution saturation and temperature. The *true induction period* is indeed hard to measure, as pointed out by Söhnel and Mullin (1978). Generally, experiments let nuclei grow to detectable size, and the time derived then is referred as *induction period*, reflecting the duration until the solid phase formation is detected (Söhnel and Garside, 1992). Hence, *induction period* is influenced by the equipment and method used to detect the formation of the solid phase in the experiments, and thus it cannot be considered a fundamental property of the system; nevertheless, analysis of values of *induction period* determined in experiments can still give some important information about the mechanism of solid phase formation and growth process (Söhnel and Mullin, 1988).

The values reported in our study represent *induction period* (and not the true induction period); they are based on experimental values, and calculated from the Avrami best fit model kinetic analysis. Considering previous studies, factors such as the temperature and surface area of the calcite sample have been shown to affect the induction period in dolomitization reactions (e.g. Sibley, 1990, Sibley et al., 1987). Hence, comparison of these experiments with other studies at different temperature and different surface area of calcite, or where different analytical methods were used, may show different induction periods. Also, Sibley et al. (1987) stated in their paper that there was considerable scatter in their data, and that they were not able to attribute this scatter to any aspect of the experimental design, but that it might have been caused by an artefact of the experimental design or due to random variations in the induction period. However, in our study, the comparison between the experiments with Zn-rich fluids and the control dolomitization experiments have the same experimental conditions of temperature, same calcite starting material, same ionic strength of the fluids, same experimental procedure, and same analytical methods, so that induction

periods in our experiments can be compared relatively to each other to assess the impact of zinc in solution on the induction period. Moreover, we have run triplicate experiments of the main experimental series in our study, which shows that our data are very consistent (and not scattered) and we have not observed any random variations in the induction period. These data give us confidence in presenting a comparison between the induction periods of the main experiments with Zn-rich fluids and the control dolomitization series in our study as derived from the Avrami models.

In terms of “when the clock starts”, we think that when we are looking at geochemical reactions, the clock would start, for example, when a mineral and fluid come into contact, or when a mineral-fluid system that was originally in equilibrium has been brought to a new stable temperature, or when two fluids are mixed, causing supersaturation with respect to a mineral. In the case of our experiments, the clock starts with bringing calcite in contact with the fluids at the reaction temperature. For natural settings, geochemical reactions may be simulated in the laboratory either through batch or flow experiments depending on the type of setting; the chemical reactions first need to be understood at the small scale, before this then can be upscaled towards regional models incorporating reactive transport models.

Concerning the role of zinc in enhancing Mg availability, the reviewer rightly points out that this process is essentially instantaneous. The formation of Mg complexes is expected to occur very quickly; we agree with this. However, this does not imply that the complex formation would decrease the induction period of the dolomitization reaction to zero. The time it takes to detect dolomite crystals will still be influenced by how fast dolomite nuclei of critical size develop and grow, impacted by temperature and supersaturation. Our experiments show clearly that (even if we were not to consider the induction period) the time from starting the experiment to reaching, for example, 50% conversion is much shorter in the saline fluids with dissolved zinc, than those without dissolved zinc.

The reviewer mentions that our data certainly raise the possibility that zinc can catalyse dolomitization reactions, and that the role zinc plays needs to be clarified a bit more explicitly as it is the major point of the paper. Firstly, we can address some concern by changing the title of the paper to “Acceleration of dolomitization by zinc in saline waters” since this reflects better what is presented in the paper. Secondly, we discuss here in more detail how the dolomitization rate of dolomitization can be affected by the chemistry of the fluids, and which role zinc may play in this.

In this discussion, we need to consider the formation of complexes (as already mentioned in the previous version of the manuscript). First, we can explain why dolomitization is expected to be faster in fluids with higher ionic strength (more saline fluids). The stability constant of metal-ligand complexes (such as magnesium complexes) decreases in fluids with higher ionic strength, following the Debye-Hückel theory and extensions. In earlier work and consistent with the results of our study (and comparison with dolomitization rates in Kaczmarek & Sibley, 2011), it has been shown that the addition of NaCl in solutions accelerates the rate of dolomitization. The addition of NaCl in the fluids means an increase in the ionic strength of the solutions. As the kinetic inhibition of dolomite formation has mainly been interpreted to relate to the strong hydration of the Mg ion (e.g. Brady et al., 1996), a decrease in the stability constant of Mg-water complexes in fluids with higher ionic strength is expected to facilitate Mg dehydration, and thus acceleration of Mg incorporation in the crystal, hence, accelerating dolomite formation. With reference to earlier work, Gaines (1974) experiments on protodolomite formation showed that the addition of LiCl also accelerates this reaction. He interpreted that dehydration of Mg ions may be a rate-controlling factor in this reaction, and that the protodolomite formation reaction rate is accelerated by the introduction of Li⁺ as this hydrated ion would reduce the water activity, and thereby help dehydration at the crystal surface accelerating crystal growth. A similar interpretation can be presented for addition of NaCl in the fluids.

For our study, we have now calculated the ratio of water molecules versus cations+anions in the solutions of our experiments to investigate further the potential impact of formation of metal-water complexes. For the solutions of our main series of experiments, we have approximately $0.8 \times N_A$ molecules of water and $0.08 \times N_A$ cations+anions (with N_A Avogadro's constant) in our 15 ml solutions. This means that we have on average 10 water molecules available per ion. Comparing hydration enthalpies of metals it is clear that the hydration enthalpy is stronger for $Zn > Mg > Ca > Na$. Previous work has shown that both Zn and Mg can form complexes with water having 6 water molecules in the first hydration shell and 12 in the second hydration shell, thus 18 water molecules attached to one cation (Bock et al., 1994; Pavlov et al., 1998; Markham et al., 2002). Furthermore, molecular dynamics simulations have shown that it is unlikely that aqueous $MgCl_2$ forms contact ion pairs (of magnesium with chloride) at ambient pressure and temperature due to the strong hydration of magnesium, and thus magnesium-water complex formation (Callahan et al., 2010). Considering the average of 10 molecules per ion, and potential of 18 water molecules around some of the cations, there may be some competitive effect in ion-water complex formation. As mentioned above, ionic strength of the solutions impacts the stability of metal-water complexes, which decreases with higher ionic strength. Also temperature impacts stability of complexes and the type of complexes. Magnesium strongly bonds with water in coordination 6, whereas Zn strongly bonds with water, but also with Cl or carbonate, and its coordination is more flexible between 4, 5 and 6 (Krezel & Maret, 2016). In Cl-rich fluids, zinc chloride complexes can form with different ratios of chloride to zinc (Maeda et al., 1996). The effect of temperature in these fluids is shown by molecular dynamics studies whereby zinc coordination of 6, the $[Zn(H_2O)_6]^{2+}$ complex, seems more stable at 25 °C, whereas the zinc coordination of 4 is more stable at 200 °C, in particular where Cl replaces one or more of the water ligands (Harris et al., 2003; Mei et al., 2015). In chloride-rich fluids, Cl can replace one or more water molecules in the complexes, and can result ultimately in $ZnCl_4^{2-}$ complexes (zinc with coordination 4, and all water molecules replaced by Cl). In biological (body temperature) context, studies indicate that the zinc binding ability is higher with biological carbonate than with chloride; there does not seem to be much research that has investigated zinc-carbonate complexation in natural (low or high temperature) fluids. In the context of our study, carbonate only comes into solution by calcite dissolution and is expected to be found at the interface of the interface coupled dissolution precipitation (dolomitization) process. The easy interconversion between 4, 5 and 6 coordination of Zn, makes zinc an effective catalyst, as shown in biological context, in particular in the zinc-enzyme carbonic anhydrase, which catalyzes the transformation of CO_2 to bicarbonate. Zinc's flexibility in coordination relates to the lack of ligand field stabilization, enabling dynamic environments of zinc ions (Krezel & Maret, 2016). Zinc differs from Mg^{2+} and Ca^{2+} because it forms much stronger complexes with water and several anions; Zn^{2+} has a high binding affinity making it a competitive towards other metal ions that bind with lower affinities. The results of our experiments that show that the addition of zinc in saline fluids accelerates the dolomitization rate seems to suggest that dehydration of the magnesium ion is helped by the zinc complex characteristics of easy interconversions between coordination and strong affinity with a variety of ligands. These characteristics may help removing water molecules from the Mg-water complex at the crystal surface by replacing Cl ligand by a water molecule on the zinc ion or changing its coordination by adding an extra water molecule.

We have clarified this role of zinc in the paper in the section lines 172-196.

Concerning the inference on supersaturation, in our experiments we use fluids with concentrations that are so high in $CaCl_2$ (0.20M) and $MgCl_2$ (0.30M) that a small amount of dissolved carbonate that comes into the fluids by dissolution of calcite is expected to trigger supersaturation with respect to dolomite. The activity of Ca^{2+} in the solutions is calculated

(using PHREEQC, for 200 °C) to be 0.0299 and the activity of Mg^{2+} 0.0727 molal. Derived from the recent work by Bénédictz et al. (2018), the solubility product for dolomite at 200 °C is calculated to be $10^{-24.01}$. This means that when we reach an activity of 2.123×10^{-11} molal for CO_3^{2-} by dissolution of calcite, we would reach saturation index of 0 with respect to dolomite. This calculation is based on saturation index with respect to dolomite = ion activity product (activity of Ca^{2+} x activity of Mg^{2+} x (activity of CO_3^{2-})²) divided by dolomite solubility product. This shows that we need only a very small amount of carbonate ions in the solution to trigger dolomite precipitation. Note that protodolomite (which forms first) has a slightly higher solubility product than stoichiometric dolomite, and hence the carbonate ion activity in the fluid would need to be a bit higher to reach protodolomite saturation in comparison with dolomite. A protodolomite solubility product at 25°C has been proposed by Hardie (1987) to be $10^{-16.52}$ which is slightly higher than the dolomite solubility product at 25°C based on the equation by Bénédictz et al. (2018) which is $10^{-17.19}$. Carbonate ions that come into solution by dissolution of calcite may be converted to bicarbonate ions based on the fairly neutral pH conditions in the solution. The calcite-to-dolomite replacement process is interpreted to be an interface-coupled dissolution precipitation process (Ruiz-Agudo et al., 2014), and at the calcite surface contact where the reaction is taking place, conversion between bicarbonate and carbonate and the rate at which this happens is hard to constrain. As the carbonate ions are not added in our fluids (in contrast to dolomite precipitation rather than calcite-to-dolomite replacement experiments) but depend on the calcite dissolution, and since the activity of the carbonate ions at the interface cannot be measured, we agree with the reviewer that it is best to remove inferences on saturation from the paper. Such statements would indeed detract from the results discussed in the paper, so we have removed this statement in our manuscript (see line 131 and 115).

References

- Bénédictz, P. Berninger, U.-N., Bovet, N., Schott, J., Oelkers, E.H. (2018) Experimental determination of the solubility product of dolomite at 50-253 °C. *Geochim. Cosmochim. Acta*, 224, 262-275.
- Bock, C.W., Kaufman, A., Glusker, J. P. (1994) Coordination of water to magnesium cations. *Inorg. Chem.*, 33, 419-427.
- Brady, P. V., Krumhansl, J. L. & Papenguth, H. W. Surface complexation clues to dolomite growth. *Geochimica Et Cosmochimica Acta* **60**, 727-731 (1996).
- Callahan, K. M., Casillas-Ituarte, N. N., Roeselová, M., Allen, H. C., Tobias, D. J. (2010) Solvation of magnesium dication: Molecular dynamics simulation and vibrational spectroscopic study of magnesium chloride in aqueous solutions. *J. Phys. Chem.*, 114, 5141-5148.
- Gaines, A. M. (1974) Protodolomite synthesis at 100 °C and atmospheric pressure, *Science*, 183, 518-520.
- Hardie, L. A. (1987) Dolomitization; a critical view of some current views. *Journal of Sedimentary Research*, 57, 166-183.
- Harris, D. J., Brodholt, J. P., Sherman, D. M. (2003) Zinc complexation in hydrothermal chloride brines: Results from ab initio molecular dynamics calculations. *J. Phys. Chem. A*, 107, 1050-1054.
- Kaczmarek, S. E., Sibley, D. F. (2011) On the evolution of dolomite stoichiometry and cation order during high-temperature synthesis experiments: An alternative model for the geochemical evolution of natural dolomites, *Sedimentary Geology*, 240, 30-40.
- Krezel, A., Maret, W. (2016) The biological inorganic chemistry of zinc ions. *Archives of Biochemistry and Biophysics*, 611, 3-19.
- Maeda, M., Ito, T., Hori, M., Johansson, G. (1996) The structure of zinc chloride complexes in aqueous solution. *Z. Naturforsch.*, 51a, 63-70.

- Markham, G. D., Glusker, J. P., Bock, C. W. (2002) The arrangement of first- and second-sphere water molecules in divalent magnesium complexes: results from molecular orbital and density functional theory and from structural crystallography. *J. Phys. Chem. B*, 106, 5118-5134.
- Mei, Y., Sherman, D. M., Liu, W., Etschmann, B., Testemale, D., Brugger, J. (2015) Zinc complexation in chloride-rich hydrothermal fluids (25-600 °C): A thermodynamic model derived from ab initio molecular dynamics. *Geochim Cosmochim Acta*, 150, 265-284.
- Pavlov, M., Siegbahn, P.E.M., Sandström, M. (1998) Hydration of beryllium, magnesium, calcium, and zinc ions using density functional theory. *J. Phys. Chem. A*, 102, 219-228.
- Ruiz-Agudo, E., Putnis, C. V., Putnis, A. (2014) Coupled dissolution and precipitation at mineral-fluid interfaces. *Chemical Geology*, 383, 132-146.
- Sibley, D. F. (1990). Unstable to stable transformations during dolomitization, *The Journal of Geology*, 98, 739–748.
- Sibley, D. F., Dedoes, R. E., Bartlett, T. R. (1987) Kinetics of dolomitization, *Geology*, 15, 1112–1114.
- Söhnel, O., Garside, J. (1992) *Precipitation*. Butterworth-Heinemann Ltd., Oxford.
- Söhnel, O., Mullin, J. W. (1978). A method for the determination of precipitation induction periods. *J. Crystal Growth*, 44, 377.
- Söhnel, O., Mullin, J. W. (1988) Interpretation of crystallization induction periods. *J. Coll. Int. Sci.*, 123, 43.

REVIEWERS' COMMENTS:

Reviewer #2 (Remarks to the Author):

I appreciate the opportunity to look at this ms. again, and am happy with their comments and their changes in this final revision. I also appreciate the authors' rebuttal comments re "induction period", and do not disagree with their use of this time datum for comparative purposes. Strictly as an aside, the reason I raised this point previously was that I think one should be careful in use of induction period in cases where observation of the actual reaction mechanism is difficult. Because the details of the transformation of calcite to dolomite are poorly understood, I simply meant that one should be cautious. Sibley has championed use of induction period as a means of "explaining" the origin of dolomite in the sedimentary record, which I think is a mistake: the term in this context yields no real insight.

Again, I appreciate being able to look at the ms. again, and congratulate the authors on a valuable contribution.

We thank the editor and referee for reviewing the revised manuscript again. In the text below, the reviewer's comments are in *italics* and our responses in non-italic letters.

Detailed reply to the reviewers' comments

Reviewers' comments:

Reviewer #2 (Remarks to the Author):

I appreciate the opportunity to look at this ms. again, and am happy with their comments and their changes in this final revision. I also appreciate the authors' rebuttal comments re "induction period", and do not disagree with their use of this time datum for comparative purposes. Strictly as an aside, the reason I raised this point previously was that I think one should be careful in use of induction period in cases where observation of the actual reaction mechanism is difficult. Because the details of the transformation of calcite to dolomite are poorly understood, I simply meant that one should be cautious. Sibley has championed use of induction period as a means of "explaining" the origin of dolomite in the sedimentary record, which I think is a mistake: the term in this context yields no real insight.

Again, I appreciate being able to look at the ms. again, and congratulate the authors on a valuable contribution.

Reply:

We thank Professor Arvidson for reviewing our manuscript again. We understand that he is happy with the comments and changes we made in the previously submitted revised version, and that he does not request further changes.

Thank you also for the additional clarification on the previous comment regarding the induction period. I agree that one should be cautious with the use of the term induction period, and that it is still useful for comparative purposes.